# Optimality-Based Non-Redfield Plankton-Ecosystem Model (OPEM v1.0) in the UVic-ESCM 2.9. Part II: Sensitivity Analysis and Model Calibration

Chia-Te Chien, Markus Pahlow, Markus Schartau, and Andreas Oschlies

GEOMAR Helmholtz Centre for Ocean Research Kiel

**Correspondence:** Chia-Te Chien (cchien@geomar.de)

**Abstract.**

We analyse 400 perturbed-parameter simulations for two configurations of an optimality-based plankton-ecosystem model (OPEM), implemented in the University of Victoria Earth-System Climate Model (UVic-ESCM), using a Latin-Hypercube sampling method for setting up the parameter ensemble. A likelihood-based metric is introduced for model assessment and selection of the model solutions closest to observed distributions of $NO_3^-$, $PO_4^{3-}$, $O_2$, and surface chlorophyll $a$ concentrations. The simulations closest to the data with respect to our metric exhibit very low rates of global $N_2$ fixation and denitrification, indicating that in order to achieve rates consistent with independent estimates, additional constraints have to be applied in the calibration process. For identifying the reference parameter sets we therefore also consider the model's ability to represent current estimates of water-column denitrification. We employ our ensemble of model solutions in a sensitivity analysis to gain insights into the importance and role of individual model parameters as well as correlations between various biogeochemical processes and tracers, such as POC export and the $NO_3^-$ inventory. Global $O_2$ varies by a factor of two and $NO_3^-$ by more than a factor of six among all simulations. Remineralisation rate is the most important parameter for $O_2$, which is also affected by the subsistence N quota of ordinary phytoplankton ($Q_{0,\,phy}^N$) and zooplankton maximum specific ingestion rate. $Q_{0,\,phy}^N$ is revealed as a major determinant of the oceanic $NO_3^-$ pool. This indicates that unraveling the driving forces of variations in phytoplankton physiology and elemental stoichiometry, which are tightly linked via $Q_{0,\,phy}^N$, is a prerequisite for understanding the marine nitrogen inventory.

## 1 Introduction

Earth system climate models (ESCMs) are powerful tools for analysing variations in climate, while resolving interdependencies between changes in the atmosphere, on land, and in the ocean (Flato, 2011; Prinn, 2013). In this regard, the dynamics of marine ecosystems is a critical link. On long timescales it regulates atmospheric $CO_2$ on the basis of biotic uptake of carbon dioxide ($CO_2$) over vast oceanic regions and due to the export of photosynthetically fixed carbon into the deep ocean, which affects the Earth's climate (Reid et al., 2009; Sigman and Boyle, 2000). Plankton ecosystem models are widely applied to understand marine biogeochemical cycles, by estimating fluxes of major elements, e.g., nitrogen, phosphorus, and carbon, as well as the

sources and sinks of marine oxygen (Maier-Reimer et al., 1995; Six and Maier-Reimer, 1996; Schmittner et al., 2005; Bopp
et al., 2013; Vallina et al., 2017; Everett et al., 2017; Ward et al., 2018).

The basic structure of most marine ecosystem models has been designed for resolving mass fluxes between nutrients, phytoplankton, zooplankton and detritus, typically referred to as NPZD models. Mathematical formulations that describe growth and fate of marine phytoplankton and zooplankton biomass have been successfully applied over a range of scales, from local 0D-ecosystem models (e.g., Fasham et al., 1990; Edwards, 2001) to global 3D models (Sarmiento et al., 1993; Keller et al., 2012; Nickelsen et al., 2015). However, most of these NPZD models lack a sound mechanistic foundation, preventing them from explicitly accounting for the organisms' regulation of their internal physiological state. For example, $N_2$ fixation by algae is often diagnosed from the availability of dissolved nutrients, so that it only occurs when the ratio of nitrate-to-phosphate concentrations falls below the Redfield ratio of 16:1 (Deutsch et al., 2007; Ilyina et al., 2013). As these assumptions neglect a number of environmental and ecological controls (e.g., grazing, often also temperature), they do not adequately describe the behaviour of plankton organisms and their sensitivity to changes in their environment. With the introduction of refined mechanistic (physiological) descriptions we here aim at alleviating this deficiency. In this study we introduce a new marine ecosystem model coupled to the University of Victoria Earth System Climate Model (UVic-ESCM, based on the configurations of Keller et al., 2012; Getzlaff and Dietze, 2013; Nickelsen et al., 2015). Doing so we anticipate the model not only to provide improved mass flux estimates, but also to exhibit more realistic sensitivities of these fluxes to varying climate conditions, e.g., in simulations of the last glacial maximum or in future projections.

In order to better represent plankton physiology, the new ecosystem model relies on optimality-based considerations for phytoplankton growth, including $N_2$ fixation (Pahlow et al., 2013; Pahlow and Oschlies, 2013), as well as zooplankton behaviour (Pahlow and Prowe, 2010). These two optimality-based models have been shown to be superior to traditional model approaches in reproducing phytoplankton and zooplankton growth and grazing under various environmental conditions (e.g., Fernández-Castro et al., 2016). Our new ecosystem model, the optimality-based plankton ecosystem model (OPEM v1.0) coupled to the UVic-ESCM, offers new features and it improves the representation of some biogeochemical properties on the global scale (e.g., net community production (NCP) and particulate C:N:P in the surface water, see Part I, Pahlow et al., 2020). One of the novel features is the representation of variable quotas of carbon (C), nitrogen (N), and phosphorus (P) in ordinary phytoplankton, diazotrophs, and particulate organic matter (detritus) exported to the deep ocean. This model approach yields mass flux estimates with spatial and temporal variations in the elemental C:N:P stoichiometry of both inorganic nutrients and organic matter as observed in situ (Loh and Bauer, 2000; Martiny et al., 2013b). PELAGOS (Vichi et al., 2007), the only ocean model with variable C:N:P in phytoplankton in CMIP5 (Bopp et al., 2013) and CMIP6 (Arora et al., 2019), has no diazotrophs, others either have only variable N:P (TOPAZ2, Dunne et al. (2013)), or variable C:P (MARBL, Danabasoglu et al., 2020). While some of the existing models have a variable C:N:P based on the optimality-based model for phytoplankton growth (Kwiatkowski et al., 2018, 2019), optimality-based $N_2$ fixation or zooplankton behaviour are not included.

Here we analyse the new model's performance and evaluate model-ensemble results against observations. Since the model is based on plankton-organism physiology, it includes new parameters whose values have not been estimated for global model applications. Also, we set up two configurations, OPEM and OPEM-H, with different temperature dependencies for diazotrophs

to investigate the effects of different empirical temperature functions on distributions of diazotrophs and $N_2$ fixation. Our analysis relies on ensembles of solutions of the two different model configurations, where every single simulation within each ensemble is subject to a different combination of parameter values. The ensembles allow assessing the sensitivity of biogeochemical tracer distributions and budgets to variations of the model's parameters. We introduce a likelihood-based metric that quantifies the global misfit between model results and observations. Amongst the ensemble simulations we regard those model solutions as the best that yield low misfits according to the metric and are also close to current estimates of water-column denitrification. The specific objectives of the present paper are (1) to identify and compare those model solutions that correspond to the best representation of observed tracer concentrations and (2) to specify the sensitivity of simulations to variations of the model's parameter values. We make inferences about the model's overall behavior, especially focusing on data constraints, limitations and advantages of resolving variable C:N:P stoichiometry for estimations of global net primary production (NPP), net community production (NCP), biogenic C export, and the global $O_2$, N, and C inventories.

## 2 Materials and Methods

### 2.1 The non-Redfield, optimality-based plankton ecosystem model in the UVic-ESCM

The optimality-based plankton ecosystem model (OPEM) has been implemented into the UVic-ESCM (Weaver et al., 2001; Eby et al., 2013), version 2.9, in the configuration of Nickelsen et al. (2015) with the isopycnal diffusivity modifications by Getzlaff and Dietze (2013), vertically increasing sinking velocity of detritus (Kriest, 2017), and several bug-fixes (some of which were already introduced by Kvale et al., 2017). The UVic-ESCM comprises three components including a simple one-layer atmospheric energy-moisture balance model (Weaver et al., 2001), a terrestrial model and a three-dimensional general ocean circulation model. The horizontal resolution of the land and ocean model components is 1.8° latitude × 3.6° longitude, and the ocean has 19 vertical levels with a thickness ranging from $50\,\mathrm{m}$ in the surface layer to $590\,\mathrm{m}$ in the deep ocean.

The OPEM and its implementation into the UVic-ESCM are described in Part I (Pahlow et al., 2020). Briefly, the major new features of the new model include (1) an optimality-based model of phytoplankton growth and diazotrophy with variable C:N:P stoichiometry (Pahlow et al., 2013), (2) the optimal current-feeding model for zooplankton (Pahlow and Prowe, 2010), and (3) variable stoichiometry in detritus. The focus on physiology in the construction of the OPEM enables us to study how biogeochemical tracer distributions and fluxes respond to different assumptions about plankton physiology.

### 2.1.1 Simulation setup

Our setup comprises ensembles of 400 simulations for each of two model configurations that differ in how temperature affects diazotrophy. The original temperature dependence of diazotrophs ($f_{\mathrm{dia}}(T)$) in the UVic-ESCM (and other models, e.g., Aumont et al., 2015), which we also employ for the OPEM configuration, limits both growth and $N_2$ fixation of diazotrophs to above $15\,°\mathrm{C}$,

$$f_{\mathrm{dia}}(T)\_\mathrm{OPEM} = \max(1.066^T - 2.6, 0)/2 \tag{1}$$

where $T$ is seawater temperature. In the OPEM-H configuration, the temperature dependence of nitrogenase activity in terrestrial systems by Houlton et al. (2008) is implemented as affecting only N$_2$ fixation,

$$f_{\text{dia}}(T)\_\text{OPEM-H} = 0.0266 * \left(1.066^T\right)^{\left[4.22 - 1.3166 * \ln\left(1.066^T\right)\right]} \tag{2}$$

while growth and nutrient uptake of diazotrophs follow the same temperature dependence as ordinary phytoplankton (see Part I, Pahlow et al., 2020). Both of these equations are empirical functions directly simulating expected or observed temperature dependencies of N$_2$ fixation. We consider Eq. (2) more realistic and hence analyse its effect on model behaviour. However, since the parameters in these two equations have no clearly identifiable physiological meaning, we consider a sensitivity analysis of the parameters in Eqs. (1) and (2) beyond the scope of the present study. Note that some models do not enforce any temperature limitation on nitrogen fixation (e.g., Dunne et al., 2013; Ilyina et al., 2013; Jickells et al., 2017). In the present ocean, waters colder than about $15\,^\circ\text{C}$ are generally replete with fixed inorganic nitrogen. For existing parameterisations of N$_2$ fixation, which are functions of the nitrate deficit with respect to phosphate, there has been little indication of substantial impacts of the formulation of temperature control at low temperatures on the distribution of nitrogen fixation (Somes and Oschlies, 2015; Landolfi et al., 2017). Such differences in formulation may, however, gain importance in environmental conditions different from today's.

For all simulations we impose preindustrial (A.D. 1850) boundary conditions with a CO$_2$ concentration of $284\,\text{ppm}$. The models have been integrated over a period of at least 10,000 years, until they reached steady-state.

The 400 parameter combinations are obtained via Latin Hypercube Sampling (LHS) (McKay et al., 1979). We vary 15 parameters in total, within the variational ranges shown in Table 1, which are based on reference ranges according to literature values. In order to reduce the number of possible parameter combinations, we vary nutrient affinities for macronutrient uptake and half-saturation concentration for iron uptake for ordinary phytoplankton and diazotrophs in constant proportions ($A_0 : A_{0,\text{D}} = 4 : 3$, $K_{Fe} : K_{\text{Fe, D}} = 1 : 2$), so that diazotrophs have a lower nutrient affinity (Pahlow et al., 2013) and higher Fe half-saturation concentration (Dutkiewicz et al., 2012; McGillicuddy Jr., 2014; Ward et al., 2013) than ordinary phytoplankton. Since our parameter sets are independent of each other, the simulations can be carried out in parallel. Apart from the computational time, the parallel setup with different parameter combinations has some advantages compared to iterative model calibration approaches, e.g., parameter-optimisation: (i) individual model simulations do not depend on any other (i.e. previous) combinations of parameter values, (ii) the ensemble results can always be re-evaluated with different metrics, perhaps with substantial differences between selected "best" solutions, depending on the error model applied, and (iii) the ensembles provide insight to the sensitivities and thus to uncertainties of particular model results with respect to parameter variations.

## 2.2 Sensitivity Analysis and Model Calibration

### 2.2.1 Sensitivity analysis

The sensitivity (Sensitivity$_T$) of a tracer $T$ to a parameter $P$ is defined here as

$$\text{Sensitivity}_T = \frac{\Delta T}{\Delta P} \times \frac{\overline{P}}{\overline{T}} \tag{3}$$

where the $\Delta$ indicates the change and the overbar the ensemble mean of $P$ or $T$. If Sensitivity$_T < 0$, the tracer and the parameter vary in opposite directions. We evaluate the sensitivities of globally and annually averaged net primary production (NPP), net community production (NCP), nitrogen fixation by diazotrophs (N$_2$ fixation), and the concentrations of oxygen (O$_2$), nitrate (NO$_3^-$), DIC, dissolved and particulate iron (DFe and PFe), Chl, ordinary phytoplankton, diazotrophs, particles (ordinary phytoplankton + diazotrophs + zooplankton + detritus) and their elemental stoichiometry to the parameters listed in Table 1. We also evaluate the sensitivities of surface particulate elemental ratios (C:N, C:P and N:P), as well as nitrate to phosphate ratios for different latitude bands (40°S to 40°N, 60°S to 70°S, and globally). This is because dissolved and particulate elemental ratios in general show very different behaviour between lower and higher latitudes (Martiny et al., 2013a). We keep all 400 simulations because we want to obtain the sensitivity information for the full parameter ranges.

**Table 1.** Parameter names, reference and variational ranges, identified "best" values for the trade-off simulations (OPEM and OPEM-H), units and descriptions. Note that the trade-off simulations share the same parameter combination.

| Symbol | Reference range | Variational range | OPEM/ OPEM-H | Units | Definition |
|---|---|---|---|---|---|
| $A_{0,\,\text{phy}}$ | 70–1000[a] | 120–280 | 229 | $\mathrm{m^3\,(mol\,C)^{-1}\,d^{-1}}$ | phytoplankton potential nutrient affinity |
| $Q_{0,\,\text{phy}}^{\text{N}}$ | 0.038–0.086[a] | 0.04–0.06 | 0.04128 | $\mathrm{mol\,(mol\,C)^{-1}}$ | phytoplankton subsistence N quota |
| $Q_{0,\,\text{dia}}^{\text{N}}$ | 0.13[a] | 0.06–0.12 | 0.067 | $\mathrm{mol\,(mol\,C)^{-1}}$ | diazotroph subsistence N quota |
| $Q_{0,\,\text{phy}}^{\text{P}}$ | 0.0008–0.002[a] | 0.0013–0.0023 | 0.0022 | $\mathrm{mol\,(mol\,C)^{-1}}$ | phytoplankton subsistence P quota |
| $Q_{0,\,\text{dia}}^{\text{P}}$ | 0.0027[a] | 0.0025–0.0035 | 0.00271 | $\mathrm{mol\,(mol\,C)^{-1}}$ | diazotroph subsistence P quota |
| $k_{\text{Fe,\,phy}}$ | 0.035–0.12[c-g] | 0.04–0.08 | 0.066 | $\mathrm{\mu mol\,m^{-3}}$ | phytoplankton half-saturation constant for Fe |
| $g_{\text{max}}$ | 0.49–5[a] | 1–2 | 1.75 | $\mathrm{d^{-1}}$ | zooplankton maximum specific ingestion rate |
| $\phi_{\text{phy}}$ | 174–765[h] | 100–200 | 118 | $\mathrm{m^3\,(mol\,C)^{-1}}$ | capture coefficient of phytoplankton |
| $\phi_{\text{dia}}$ | $1.05 \cdot \phi_{\text{phy}}$[i] | 150–250 | 232 | $\mathrm{m^3\,(mol\,C)^{-1}}$ | capture coefficient of diazotrophs |
| $\phi_{\text{det}}$ | $\phi_{\text{phy}}$[c-f] | 20–100 | 94 | $\mathrm{m^3\,(mol\,C)^{-1}}$ | capture coefficient of detritus |
| $\phi_{\text{zoo}}$ | 0–3230[h] | 100–200 | 118 | $\mathrm{m^3\,(mol\,C)^{-1}}$ | capture coefficient of zooplankton |
| $\lambda_{0,\,\text{phy}} = M_{0,\,\text{dia}}$ | 0.001–0.015[c-f] | 0.01–0.03 | 0.018 | $\mathrm{d^{-1}}$ | specific mortality rate |
| $\nu_{\text{det}}$ | 0.05–0.07[c-g] | 0.04–0.09 | 0.087 | $\mathrm{d^{-1}}$ | remineralization rate |

[a](Pahlow, 2005; Pahlow et al., 2013), [b](Pahlow and Prowe, 2010), [c](Keller et al., 2012), [d](Somes and Oschlies, 2015), [e](Somes et al., 2017) [f](Landolfi et al., 2017), [g](Landolfi et al., 2015), [h](Su et al., 2018), [i](Wang et al., 2019)

### 2.2.2 Likelihood-based metric assessing global biogeochemical model results

We consider four different types of observations for quantitatively assessing the model simulations. The first three are the objectively analysed monthly (upper 550 m) and annual (below 550 m) concentrations of nitrate, phosphate, and oxygen of the World Ocean Atlas 2013 (WOA 2013, Garcia et al., 2013a, b). The fourth is the monthly mean chlorophyll concentration derived from remote sensing data (MODIS/Aqua level 3), based on monthly climatologies for 10 years from 2008 to 2017,

provided by the ocean biology processing group (Ocean Biology Processing Group, 2014). The satellite-derived chlorophyll (Chl) concentrations are used for data-model comparison only for the UVic model's top layer, i.e. the upper $50\,\mathrm{m}$.

We define our metric in terms of spatial averages of 17 distinct biogeochemical biomes, as derived and described by Fay and McKinley (2014). The individual biomes are regarded as regions of common biogeochemistry and thus account for spatial differences between ocean regions on the largest possible (global) scale. Using 56 biogeochemical provinces, as defined by Longhurst (2007), might have hampered our data-model comparison, because a higher resolution of individual regions can accentuate spatial pattern errors in tracer concentrations, resulting from model errors in advection and mixing. In our view the biomes of Fay and McKinley (2014) are coarse enough for avoiding this problem, but still sufficiently informative for identifying representative parameter values.

The underlying error model of the likelihood based metric assumes a Gaussian (normal) distribution, which is well represented by using the first two moments of log-transformed tracer concentrations, in particular for the upper ocean layers (Schartau et al., 2017). For every depth-level of the UVic model ($k \in \{1, 2, 3, \ldots, 19\}$), average $\log_{10}$-transformed tracer concentrations ($\overline{\log_{10}\mathrm{X}}$) of type X are determined as spatial arithmetic means for our 17 biomes (indexed as $j$ in Eq. 4) for the observations and model results:

$$\left(\overline{\log_{10}\mathrm{X}}\right)_{jk} = \frac{1}{\mathrm{N}_{jk}} \sum_{n=1}^{\mathrm{N}_{jk}} \left(\log_{10}\left[\frac{\max(\mathrm{X}_{(n)}, \mathrm{X}_{(0)})}{\mathrm{X}_{(0)}}\right]\right), \qquad \mathrm{X} \in \{\mathrm{Chl}, \mathrm{O}_2, \mathrm{NO}_3{}^-, \mathrm{PO}_4{}^{3-}\} \tag{4}$$

where $\mathrm{N}_{jk}$ is the number of available data points within biome $j$ in depth level $k$. Prior to the $\log_{10}$-transformation, all tracer concentrations have been normalised to lower detection (uncertainty) thresholds ($\mathrm{X}_{(0)}$) respectively. Measured or derived concentrations below these thresholds are treated as noise and therefore remain unresolved. Thus, the $\log_{10}$-transformed normalised concentrations are non-negative. The threshold-values are: $\mathrm{Chl}_{(0)} = 0.1\,\mathrm{mg\,m^{-3}}$, $\mathrm{O}_{2(0)} = 1\,\mathrm{mmol\,m^{-3}}$, $\mathrm{NO}_3{}^-{}_{(0)} = 0.05\,\mathrm{mmol\,m^{-3}}$, and $\mathrm{PO}_4{}^{3-}{}_{(0)} = 0.01\,\mathrm{mmol\,m^{-3}}$.

Our metric is derived from a likelihood, assuming a Gaussian error distribution for the residuals, which describe the discrepancy between mean values derived from observations ($\overline{\log_{10}\mathrm{X}^{(\mathrm{obs})}}$) and model simulations ($\overline{\log_{10}\mathrm{X}^{(\mathrm{mod})}}$). Hereafter we refer to this metric as our cost function ($J$). Our cost function is split up into two major parts:

$$J = \sum_{k=1}^{5} J_k^{(u)} + \sum_{k=6}^{19} J_k^{(l)} \tag{5}$$

$$J_k^{(u)} = \sum_{i=1}^{12} \sum_{j=1}^{17} \left[\mathbf{d}^T R^{-1} \mathbf{d}\right]_{ijk} + \left(\mathbf{v}^{(\mathrm{obs})} - \mathbf{v}^{(\mathrm{mod})}\right)_{ijk}^T V_{ijk}^{-1} \left(\mathbf{v}^{(\mathrm{obs})} - \mathbf{v}^{(\mathrm{mod})}\right)_{ijk} \tag{6}$$

$$J_k^{(l)} = \sum_{j=1}^{17} \left[\mathbf{d}^T R^{-1} \mathbf{d}\right]_{jk} + \left(\mathbf{v}^{(\mathrm{obs})} - \mathbf{v}^{(\mathrm{mod})}\right)_{jk}^T V_{jk}^{-1} \left(\mathbf{v}^{(\mathrm{obs})} - \mathbf{v}^{(\mathrm{mod})}\right)_{jk} \tag{7}$$

where $\mathbf{d}$ is the residual vector (see Eq. (8) below), $R$ the covariance matrix (Eq. 9), $\mathbf{v}^{(\mathrm{obs})}$ and $\mathbf{v}^{(\mathrm{mod})}$ the spatial variance estimates of the $\log_{10}$-transformed observed and modelled tracers, and $V^{-1}$ are diagonal matrices with the variances (uncertainties) of $\mathbf{v}^{(\mathrm{obs})}$. The first part ($J_k^{(u)}$) of the cost function resolves seasonal changes between the surface and $550\,\mathrm{m}$ depth,

corresponding to the upper five depth-levels of the model. The second part ($J_k^{(l)}$) represents the lower depth range below $550\,\mathrm{m}$ and does not account for seasonal changes, as only annual mean data are available.

The residual vector ($\mathbf{d}$) (whose components represent the tracer types X) used for $J$ describes the differences between the $\log_{10}$-transformed observations and their model counterparts:

$$\mathbf{d}_{ijk} = \left( \overline{\log_{10} \mathbf{X}_{ijk}^{(\mathrm{obs})}} - \overline{\log_{10} \mathbf{X}_{ijk}^{(\mathrm{mod})}} \right) \tag{8}$$

where $i$ and $j$ are the month and biome indices, respectively. We recall that $\mathbf{d}$ has four components only for the UVic model's top layer ($k = 1$) where chlorophyll data are regarded as well. For $k > 1$ the residual vector contains three components: $O_2$, $NO_3^-$, and $PO_4^{3-}$. Both parts of the cost function ($J_k^{(u)}$ and $J_k^{(l)}$) in turn contain two terms, one with respect to the residuals, as defined in Eq. (8), and another that accounts for the differences between the spatial variances (vectors $\mathbf{v}_{ijk}^{(\mathrm{obs})}$ and $\mathbf{v}_{ijk}^{(\mathrm{mod})}$) within each biome (and month for $J_k^{(u)}$) at each depth-level. The covariance matrices $R_{ijk}$ account for temporal correlations ($C_{jk}$) between different variables ($\mathbf{X}^{(\mathrm{obs})}$), that are specified for every biome and depth level separately:

$$R_{ijk} = S_{ijk} \cdot C_{jk} \cdot S_{ijk} \tag{9}$$

where the elements of the diagonal matrices $S_{ijk}$ are the standard errors of the mean $\log_{10}$-transformed tracer concentrations ($\overline{\log_{10} \mathbf{X}_{ijk}^{(\mathrm{obs})}}$) calculated in Eq. (4) for every month $i$, biome $j$, and depth level $k$. For $J_k^{(l)}$ the $R_{jk}$ contain only the squared standard errors of the annual data as diagonal elements ($R_{jk} = S_{jk}^2$).

With the consideration of standard errors instead of standard deviations, we implicitly impose weights to differences in the spatial expansion (i.e. number of data points of the gridded product used) of individual biomes. Overall, the final cost function $J$ resolves spatial differences between regions (biomes) as well as temporal differences for those depth levels where monthly data are available. It is thus a combination of time-varying and spatial information for the assessment of our biogeochemical model results on a global scale.

In order to estimate uncertainty ranges for selected model results (globally-averaged $N_2$ fixation, $NO_3^-$, $O_2$, DIC concentrations, NPP, NCP), we apply a bootstrap method to obtain an uncertainty quantification for our simulated values based on the 400 available ensemble model simulations. We collect the best solutions (lowest cost function value) of 1000 randomly selected subsets of 100 out of our 400 ensemble members. The mean and 95% confidence interval of these subsets provide an uncertainty range in the vicinity of the value of the full ensemble.

## 3 Results

Table 2 lists the ranges of selected simulated tracers and processes for the full ensemble of parameter values generated by the Latin Hypercube Sampling for the OPEM and OPEM-H configurations. Our results exhibit wide ranges of tracer concentrations and fluxes in these two configurations. In particular, globally-averaged $NO_3^-$ concentrations range from 10.2 to $66.2\,\mathrm{mmol\,m^{-3}}$ and integrated $N_2$ fixation from 0 to $518\,\mathrm{Tg\,N\,yr^{-1}}$. Tracers in OPEM and OPEM-H show similar ranges, except for globally averaged $NO_3^-$, which ranges from 10.2 to $66.2\,\mathrm{mmol\,m^{-3}}$ in OPEM and 13.0 to $55.0\,\mathrm{mmol\,m^{-3}}$ in OPEM-H.

**Table 2.** Ranges of global averages of major tracer concentrations or fluxes in the OPEM and OPEM-H configurations. Chl concentration is for the upper 50 m (surface layer of the UVic grid) and NCP is for the upper 100 m. Observations and reference model simulations are listed in the Reference column.

| Tracer | OPEM | OPEM-H | Reference | Units |
|--------|------|--------|-----------|-------|
| Oxygen | 99.6–219 | 103–214 | 176[a] | $\mathrm{mmol\,m^{-3}}$ |
| Nitrate | 10.2–66.2 | 13.0–55.0 | 31[b] | $\mathrm{mmol\,m^{-3}}$ |
| DIC | 2.239–2.439 | 2.248–2.430 | 2.317[c] | $\mathrm{mol\,m^{-3}}$ |
| DFe | 0.47–0.71 | 0.47–0.69 | 0.57[d] | $\mathrm{\mu mol\,m^{-3}}$ |
| PFe | 0.44–0.75 | 0.44–0.70 | 1.17[d] | $\mathrm{nmol\,m^{-3}}$ |
| Chl | 0.123–0.332 | 0.128–0.336 | 0.309[e] | $\mathrm{mg\,m^{-3}}$ |
| NPP | 27.8–88.0 | 27.2–88.0 | 52[f] | $\mathrm{Pg\,C\,yr^{-1}}$ |
| NCP | 8.0–16.4 | 7.8–16.2 | 13.5[g] | $\mathrm{Pg\,C\,yr^{-1}}$ |
| $N_2$ Fixation | 0–480 | 0–518 | 140[h] | $\mathrm{Tg\,N\,yr^{-1}}$ |

[a] WOA 2013 (Garcia et al., 2013a)

[b] WOA 2013 (Garcia et al., 2013b)

[c] GLODAPv2 (Olsen et al., 2016)

[d] (Nickelsen et al., 2015),

[e] MODIS/Aqua level 3, 2008–2017 (Ocean Biology Processing Group, 2014)

[f] (Westberry et al., 2008)

[g] (Li and Cassar, 2016)

[h] (Luo et al., 2012)

## 3.1 Sensitivity to Model Parameters

### 3.1.1 Biogeochemical tracer inventories and governing processes

The sensitivities of globally averaged biogeochemical properties to the variations of each of the 13 parameters in Table 2 are comparable for OPEM and OPEM-H (Figure 1). Global mean oxygen concentration is most sensitive to $\nu_{\mathrm{det}}$ (remineralization rate). Higher $\nu_{\mathrm{det}}$ increases oxygen consumption in shallow water, where oxygen resupply from the atmosphere is stronger. Less oxygen is consumed below the surface ocean, hence the total oxygen inventory increases. Maximum ingestion rate ($g_{\mathrm{max}}$) and grazing rate on ordinary phytoplankton ($\phi_{\mathrm{phy}}$) also correlate positively with oxygen. Higher $g_{\mathrm{max}}$ or $\phi_{\mathrm{phy}}$ means more ordinary phytoplankton is grazed and less particles are formed, which then decreases oxygen consumption through remineralization. Oxygen is less sensitive to $\phi_{\mathrm{dia}}$, because the biomass of diazotrophs is much smaller than that of ordinary phytoplankton.

A surprising finding is that oxygen is sensitive to, and positively correlated with, the subsistence nitrogen quota of ordinary phytoplankton ($Q_{0,\,\mathrm{phy}}^{\mathrm{N}}$). From a classic point of view, oxygen levels in the ocean are dominated by physical supply processes as well as biogeochemical consumption processes such as remineralization (Feely et al., 2004). Nevertheless, in our simulations

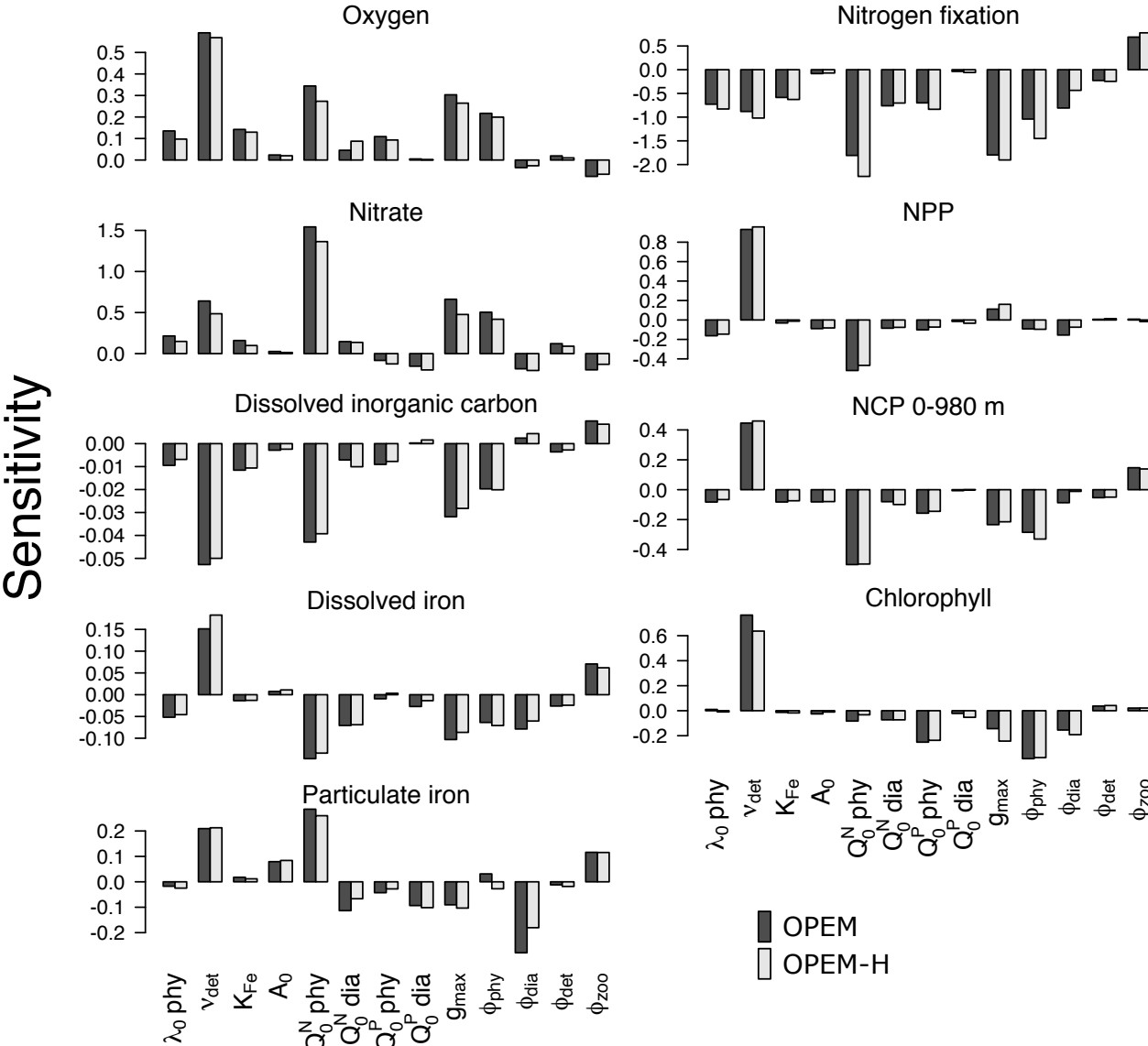

**Figure 1.** Sensitivities of globally averaged $O_2$, $NO_3^-$, dissolved inorganic carbon, dissolved iron, particulate iron, $N_2$ fixation, net primary production (NPP), Chlorophyll, and net community production (NCP) integrated from 0 to 980 m to individual model parameters, computed according to Eq. (3). Note the different y-axis ranges in the different panels.

the sensitivity to $Q_{0,\,phy}^N$ is more than half (58%) of that to $\nu_{det}$ in OPEM and 48% in OPEM-H (Figure 1). In our model, $Q_{0,\,phy}^N$ has no effect on the spatial distribution of cellular C:N ratios in phytoplankton, which is determined by ambient light and nutrient conditions. However, $Q_{0,\,phy}^N$ affects the average phytoplankton C:N ratio. The average phytoplankton C:N ratio decreases when $Q_{0,\,phy}^N$ increases, with less carbon being fixed for the same $NO_3^-$ supply. Oxygen consumption (due to remineralization)

per mole of nitrogen thus decreases in consequence. $Q_{0,\,\mathrm{phy}}^{\mathrm{N}}$ in turn affects $\mathrm{NO_3}^-$: A higher $Q_{0,\,\mathrm{phy}}^{\mathrm{N}}$ yields a higher oxygen level and hence less denitrification in oxygen deficient zones (ODZs) and therefore leads to more $\mathrm{NO_3}^-$. In fact, we identify this as a major process that controls the $\mathrm{NO_3}^-$ inventory in our simulations (Figure 1). While $\mathrm{NO_3}^-$ is also sensitive to other parameters, its sensitivity to $Q_{0,\,\mathrm{phy}}^{\mathrm{N}}$ is more than twice that to any other parameter (Figure 1).

The sensitivity of dissolved inorganic carbon (DIC) is generally low, because of the relatively large DIC pool compared to the variations in fluxes among the different parameter sets. Similar to oxygen, DIC is most sensitive to $\nu_{\mathrm{det}}$, $Q_{0,\,\mathrm{phy}}^{\mathrm{N}}$, $g_{\mathrm{max}}$ and $\phi_{\mathrm{phy}}$. Faster carbon recycling in the surface layer due to higher $\nu_{\mathrm{det}}$ generates a higher surface DIC concentration and hence more outgassing, which decreases the DIC inventory. A somewhat lower DIC inventory is also induced by a larger $Q_{0,\,\mathrm{phy}}^{\mathrm{N}}$, as less carbon is fixed and exported per unit nitrogen in phytoplankton, and by enhanced zooplankton grazing with larger $g_{\mathrm{max}}$.

Dissolved iron (DFe) is most sensitive to the remineralisation rate ($\nu_{\mathrm{det}}$). Unlike $\mathrm{NO_3}^-$, which has dynamic source ($\mathrm{N_2}$ fixation) and sink (denitrification) processes, iron has a fixed source from atmospheric deposition and a sink in the sediment, and the size of the DFe pool is mainly determined by its internal cycle. A higher remineralisation rate prolongs the residence time and thus increases the DFe pool. The parameter $\nu_{\mathrm{det}}$ also indirectly affects the internal DFe cycle via its effect on $\mathrm{O_2}$. While the detritus remineralisation rate drops when $\mathrm{O_2}$ falls below $5\,\mathrm{mmol\,m^{-3}}$ (Nickelsen et al., 2015), scavenging of DFe stops below the same oxygen threshold. Detritus remineralisation rate dominates variations in DFe when globally averaged $\mathrm{O_2}$ is above $135\,\mathrm{mmol\,m^{-3}}$, in which case DFe is positively correlated with $\nu_{\mathrm{det}}$ and $\mathrm{O_2}$. When globally averaged $\mathrm{O_2}$ is below $135\,\mathrm{mmol\,m^{-3}}$, the wide-spread ODZs (below $5\,\mathrm{mmol\,m^{-3}}$) inhibit the scavenging of DFe and this effect dominates. As a result, DFe becomes anti-correlated with $\mathrm{O_2}$. Particulate iron (PFe) is also positively correlated with $\nu_{\mathrm{det}}$ when globally averaged $\mathrm{O_2}$ is above $135\,\mathrm{mmol\,m^{-3}}$, but below that PFe shows no correlation with $\nu_{\mathrm{det}}$. When globally averaged $\mathrm{O_2}$ is below $135\,\mathrm{mmol\,m^{-3}}$, inhibition of scavenging of DFe in ODZs decreases PFe there but a higher DFe increases PFe elsewhere, because PFe is coupled to DFe through scavenging and remineralisation. As mentioned above, $Q_{0,\,\mathrm{phy}}^{\mathrm{N}}$ controls the average nitrogen quota in phytoplankton and thus in particles. Since PFe is proportional to the amount of nitrogen in particles, $Q_{0,\,\mathrm{phy}}^{\mathrm{N}}$ also affects PFe. This (positive) sensitivity is much stronger than the indirect (negative) effect via DFe leading to opposite sensitivities of DFe and PFe to $Q_{0,\,\mathrm{phy}}^{\mathrm{N}}$. Other than $\nu_{\mathrm{det}}$ and $Q_{0,\,\mathrm{phy}}^{\mathrm{N}}$, PFe is also sensitive to $\phi_{\mathrm{dia}}$ because dead diazotrophs enter the particulate pool (detritus) and diazotrophs are very sensitive to $\phi_{\mathrm{dia}}$ (Figure 2).

The simulated global $\mathrm{N_2}$ fixation rate is sensitive to many parameters, apart from $A_{0,\,\mathrm{phy}}$ and $Q_{0,\,\mathrm{dia}}^{P}$. Similar relative changes in most parameters introduce changes to the global $\mathrm{N_2}$ fixation rate that are of similar magnitude. Interestingly, $\mathrm{N_2}$ fixation is sensitive also to zooplankton parameters, indicating that zooplankton grazing on diazotrophs is an important factor controlling not just diazotroph biomass but also $\mathrm{N_2}$ fixation.

Of particular interest are the sensitivities of global net primary production (NPP) and net community production (NCP). Particle fluxes in marine biogeochemical models tend to agree most closely with sediment trap data for depths of about $1000\,\mathrm{m}$ or below (Kriest et al., 2012). Therefore, different from Table 2, showing NCP for the upper $100\,\mathrm{m}$ for comparison with observations and other (reference) model simulations, here we integrate NCP from 0 to $980\,\mathrm{m}$ ($7^{\mathrm{th}}$ layer of the ocean in the UVic-ESCM), which in steady state is equivalent to POC export flux at $980\,\mathrm{m}$. NPP is sensitive to $\nu_{\mathrm{det}}$ and $Q_{0,\,\mathrm{phy}}^{\mathrm{N}}$. A higher $\nu_{\mathrm{det}}$ causes faster nutrient recycling in surface waters, which increases NPP and reduces particle export and hence NCP. Increasing

$Q_{0,\,phy}^{N}$ lowers both NPP and NCP and hence also the fixed-carbon inventory. A higher ingestion rate of zooplankton ($g_{max}$) removes more particles and thus is negatively correlated with NCP. Chl is the principal agent of C fixation in the OPEM and hence Chl has a similar sensitivity pattern as NPP except for $g_{max}$ and $\phi_{phy}$.

### 3.1.2 Ordinary phytoplankton, diazotrophs, particles, export and their elemental stoichiometry

First we discuss the proportions of carbon, nitrogen and phosphorus in ordinary phytoplankton and diazotrophs, since variations in elemental stoichiometry in autotrophs originate in differential uptake of nutrients under different environmental conditions.

Globally averaged C, N, P concentrations and ratios of globally averaged N and P of ordinary phytoplankton and diazotrophs are sensitive to $\nu_{det}$, $Q_{0,\,phy}^{N}$, $\phi_{phy}$ and $\phi_{dia}$ (Figure 2). As expected, C, N and P of ordinary phytoplankton and diazotrophs increase for higher $\nu_{det}$, which generates higher nutrient concentrations in the surface ocean. They are also sensitive to zooplankton grazing, especially to $\phi_{phy}$ and $\phi_{dia}$. $Q_{0,\,phy}^{N}$ and $Q_{0,\,phy}^{P}$ are negatively correlated with ordinary phytoplankton C, indicating that the negative effect of higher subsistence quotas on competitive ability dominates their effect on biomass. A similar behavior is found in diazotrophs except that $Q_{0,\,dia}^{N}$ is also negatively correlated with diazotroph N and hence also nitrogen fixation (Figure 1). Although an increase in $Q_{0,\,phy}^{N}$ makes ordinary phytoplankton less competitive, it also raises the oceanic $NO_3^{-}$ inventory, which eventually leads to more phytoplankton N (Figure 2) and less nitrogen fixation (Figure 1).

Diazotroph C, N and P are generally more sensitive to parameter variations than phytoplankton, due to the much smaller total biomass of diazotrophs, which is also the reason why diazotrophs are less sensitive in OPEM-H, the model configuration in which their biomass is generally larger because of the growth of diazotrophs at high latitudes (see Fig. 15 in Part I, Pahlow et al., 2020). Since ordinary phytoplankton dominates autotrophic biomass, it tends to control nutrient distributions. This explains why ordinary phytoplankton parameters such as $Q_{0,\,phy}^{N}$ and $\phi_{phy}$ have strong effects on diazotrophs but not vice versa. The zooplankton grazing preferences $\phi_{phy}$ and $\phi_{dia}$ drive the competition between ordinary phytoplankton and diazotrophs and hence have strong and opposing effects on their biomass. Owing to the relatively small total biomass, diazotroph C is more sensitive to changes in $\phi_{phy}$ and $\phi_{dia}$ than ordinary phytoplankton C.

Particulate C:N and N:P ratios are most sensitive to $Q_{0,\,phy}^{N}$ (Figure 3). This sensitivity is related to biomass, as we see from the OPEM-H configuration, where (non-$N_2$ fixing) diazotrophs are abundant at high latitudes (see Fig. 15 in Part I, Pahlow et al., 2020) and consequently the sensitivity of high-latitude C:N to $Q_{0,\,dia}^{N}$ is high, even higher than to $Q_{0,\,phy}^{N}$ (Figure 3). We do not find this behavior for high-latitude regions in the OPEM configuration, as well as low-latitude regions, where diazotrophs are not as abundant. The parameter $Q_{0,\,phy}^{P}$ was expected to be the most important parameter for particulate C:P ratios, just like $Q_{0,\,phy}^{N}$ is for the C:N ratio. However, this is only true for the OPEM at high latitudes.

At low latitudes, particulate C:P ratios are most sensitive to $Q_{0,\,phy}^{N}$ (Figure 3). The supply of nitrate and phosphate at different latitudes is the major reason for this pattern. At low latitudes, the effects of $Q_{0,\,phy}^{P}$ are suppressed by variations in phytoplankton C, which is affected by $Q_{0,\,phy}^{N}$ and the consequent change in nitrate concentration. Nitrate and phosphate are not limiting in the high-latitude Southern Ocean where, under N- and P-replete conditions, cellular C:P is mainly determined by $Q_{0,\,phy}^{P}$ and a higher $Q_{0,\,phy}^{P}$ would result in a higher cellular P:C (lower C:P). Therefore, the global C:P of total particulate matter, which is dominated by ordinary phytoplankton, is negatively correlated with $Q_{0,\,phy}^{P}$.

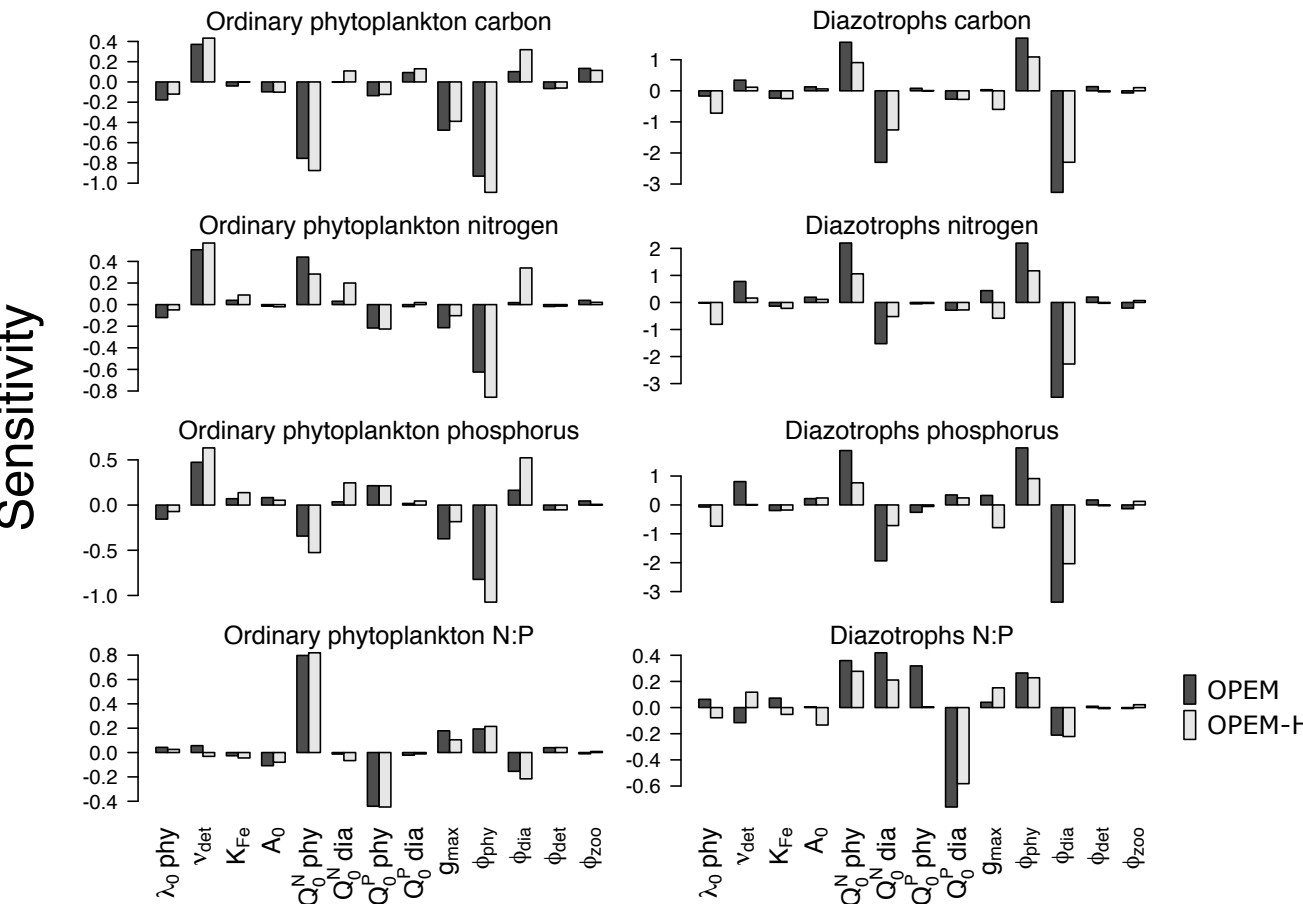

**Figure 2.** Parameter sensitivities of globally averaged concentrations of ordinary phytoplankton and diazotrophs carbon, nitrogen, phosphorus, and ratios of globally averaged N and P. Black and grey shading denote OPEM and OPEM-H configurations, respectively. Note the different y-axis ranges in the different panels.

The sensitivities of dissolved N:P ratio to parameters in the three geographical settings (low, high latitudes and global) follow similar patterns. However, we find sensitivities to be generally higher in the low-latitudes, especially to variations of the phytoplankton parameters. Again this is because $NO_3^-$ is often limiting in lower latitudes, particularly in the oligotrophic gyres, where the dissolved nitrogen pool is more sensitive to changes in phytoplankton as well as $N_2$ fixation. This is also why grazing pressure on diazotrophs ($\phi_{dia}$) has a much stronger effect at low than at high latitudes.

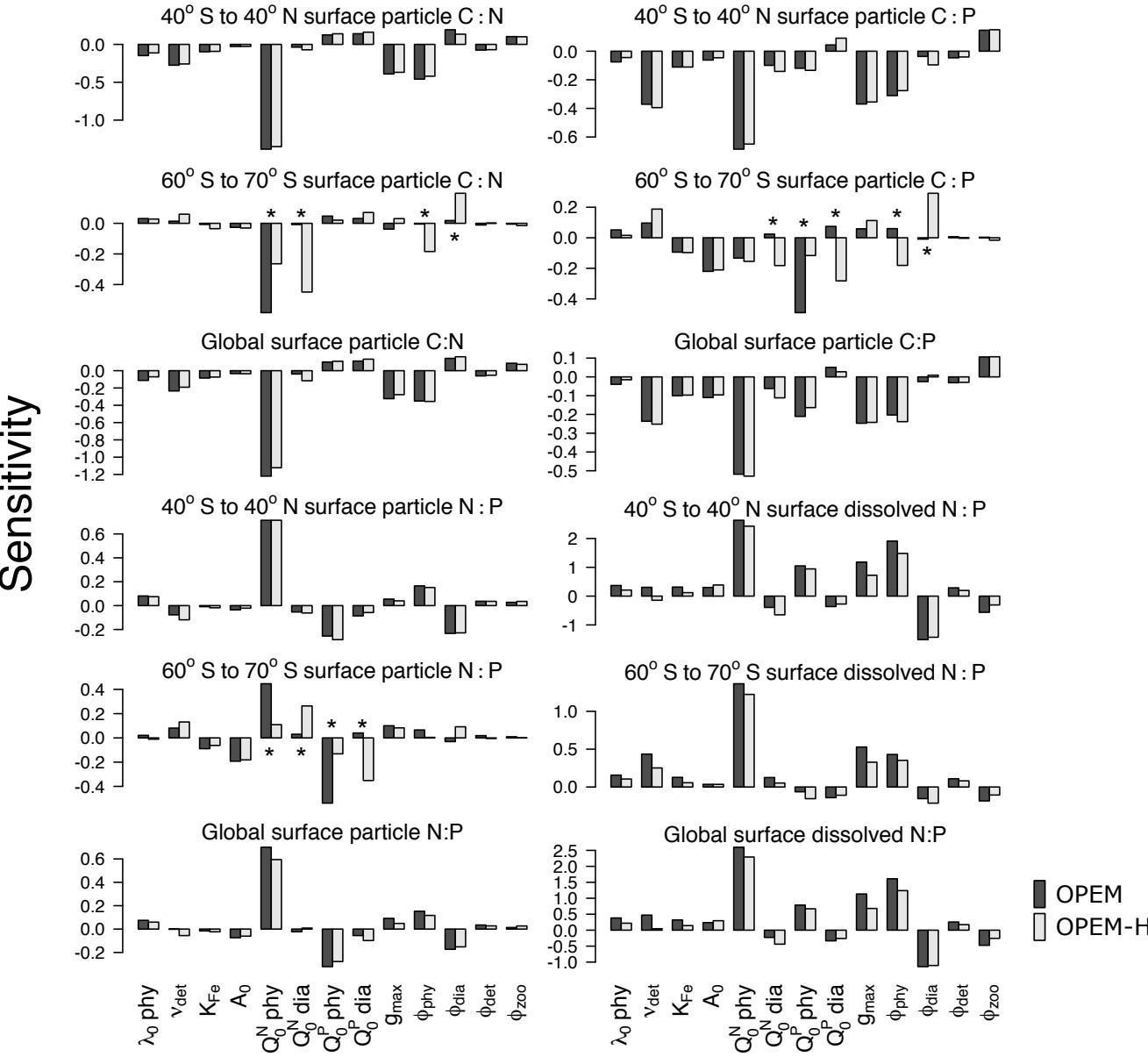

**Figure 3.** Parameter sensitivities of averaged surface (0–130 m) particulate elemental C:N, C:P, and N:P ratios for different latitude bands (40°S to 40°N, 60°S to 70°S, and the global ocean). Asterisks indicate sensitivities that are very different between OPEM and OPEM-H. Note the different y-axis ranges in the different panels.

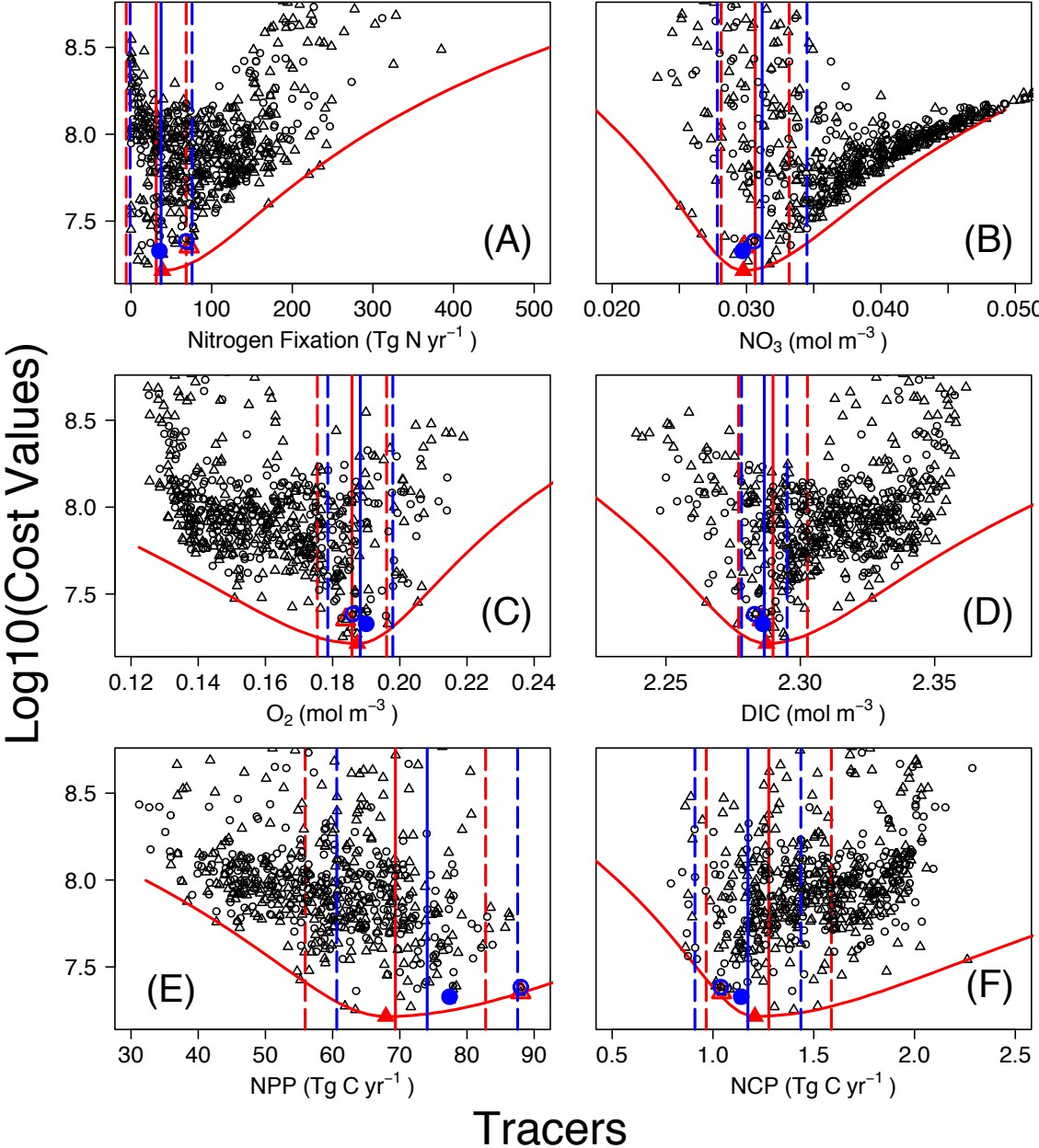

**Figure 4.** Costs vs. tracer concentrations and fluxes for annual $N_2$ fixation (A), globally averaged $NO_3^-$ (B), $O_2$ (C) and dissolved inorganic carbon (DIC) (D) concentrations, as well as annual net primary production (NPP) (E) and net community production (NCP, here integrated over the depth range 0 to 980 m) (F). Red and blue symbols and lines are for OPEM (triangles) and OPEM-H (circles), respectively. Solid and open symbols represent minimum-cost and trade-off simulations, respectively. Vertical solid and dashed lines represent mean and 95% confidence interval of best solutions of 1000 randomly selected subsets of 100 ensemble members. Red parabolas fit the lowest costs at different rates or tracer concentrations.

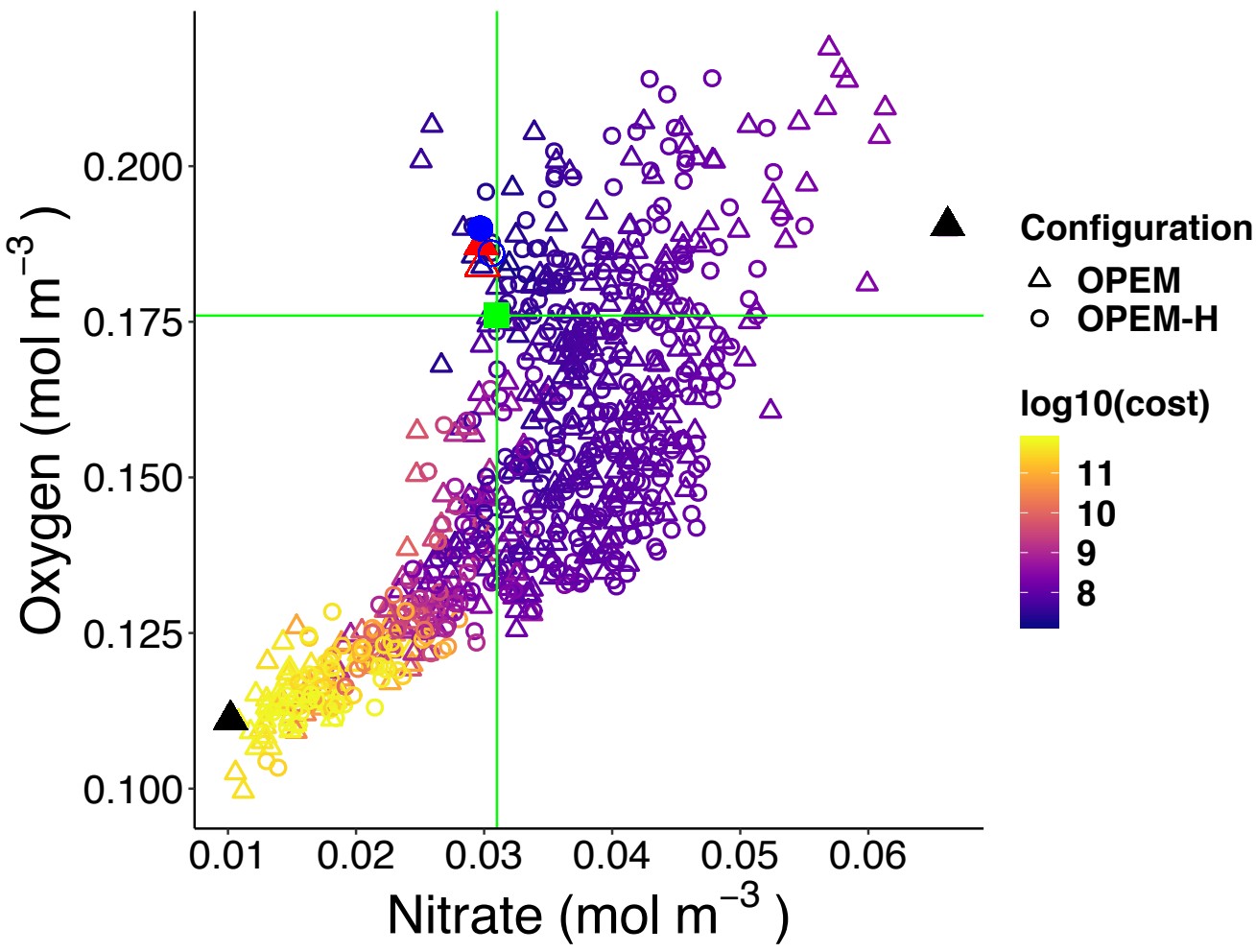

**Figure 5.** Globally averaged oxygen vs. nitrate in OPEM and OPEM-H. Color represents cost value. Solid red triangle and blue circle annotate the simulations with minimum cost in OPEM and OPEM-H, respectively, and open red triangle and blue circle are the trade-off simulations. The green square, horizontal and vertical lines indicate mean oxygen and nitrate concentrations of $0.176\,\mathrm{mol\,m^{-3}}$ and $0.031\,\mathrm{mol\,m^{-3}}$, respectively, in the WOA 2013. Solid black triangles highlight the lowest and highest $NO_3^-$ simulations used in Figure 6 and 7.

## 3.2 Cost function values of the ensemble simulations

### 3.2.1 Constraining global rate estimates and inventories

The cost function (introduced in Section 2.2.2) was devised for identifying the best solutions among the ensemble runs. For the model's upper layers ($0 - 550\,\mathrm{m}$) observational monthly mean concentrations of nitrate and phosphate enter the cost function, thereby reflecting regional and seasonal variations in the N:P uptake ratio of ordinary phytoplankton and diazotrophs. Variations in nitrate and phosphate availability affect the growth of diazotrophs and thus determine global $N_2$ fixation in both OPEM and OPEM-H. In our UVic configurations, water column denitrification is the only fixed-N loss term. Therefore, the simulated $N_2$ fixation is expected to match water column denitrification under a steady-state nitrogen cycle. Nevertheless, the simulation with the lowest cost yields a global $N_2$-fixation rate estimate of $38.8\,\mathrm{Tg\,N\,year}^{-1}$ (Figure 4A), much lower than recent estimates of water column denitrification ($55.8$ - $72.9\,\mathrm{Tg\,N\,year}^{-1}$; Somes et al., 2017; Wang et al., 2019).

The cost function penalises solutions that yield $N_2$ fixation rates greater than $90\,\mathrm{Tg\,N\,year}^{-1}$, but shows no clear relation to $N_2$ fixation at lower rates (Figure 4A). For example, among the simulations with the 5 lowest cost function values in the OPEM configuration, the global ocean $N_2$ fixation rate varies between 8 and $40\,\mathrm{Tg\,N\,year}^{-1}$. These model solutions also differ with respect to their $O_2$ inventories. The tendency of the cost function to favor very low global $N_2$ fixation is caused by a compensatory effect, whereby improving $NO_3^-$ deteriorates $O_2$ and vice versa (see also Part I, Pahlow et al., 2020, and the Discussion section below). Thus, instead of selecting the reference parameter sets based only on the cost function, we also take the ability to yield reasonable $N_2$ fixation rates into account, whereby we ignore simulations with rates below $60\,\mathrm{Tg\,N\,year}^{-1}$, since this is the lower boundary of current data-based estimates of water-column denitrification (DeVries et al., 2012). As these solutions represent a somewhat subjective trade-off between low cost and reasonable $N_2$ fixation, we refer to them as trade-off solutions and details of their behaviour are shown and discussed in Part I (reference simulations in Pahlow et al., 2020). For OPEM the trade-off solution corresponds to the seventh-lowest cost function value, and the fourth-lowest for OPEM-H.

In the following we will describe the lowest-cost solutions together with the trade-off solutions, as well as respective uncertainty ranges obtained from the bootstrap method described in the Materials and Methods section. The width of the uncertainty ranges (95% confidence intervals) in Figure 4 indicates the metric's ability to constrain the inventory or rate under consideration. Globally averaged $N_2$ fixation rates of our trade-off solutions of OPEM and OPEM-H are just outside and within this uncertainty range, respectively (Figure 4A). The global $NO_3^-$ inventory turns out to be remarkably well constrained (Figure 4B). The mean global estimates are $30.6\,\mathrm{mmol\,N\,m}^{-3}$ and $31.4\,\mathrm{mmol\,N\,m}^{-3}$ for OPEM and OPEM-H, respectively. Ensemble solutions that deviate from these estimates have high costs and therefore the uncertainty ranges remain narrow. The trade-off and minimum-cost solutions are hardly distinguishable. The uncertainty of the simulated global $O_2$ is comparable to that of the $NO_3^-$ inventory. Global mean $O_2$ concentrations of OPEM and OPEM-H are $186\,\mathrm{mmol\,O_2\,m}^{-3}$ and $187\,\mathrm{mmol\,O_2\,m}^{-3}$. Our metric effectively constrains global DIC estimates, $2.290\,\mathrm{mol\,C\,m}^{-3}$ for OPEM and $2.287\,\mathrm{mol\,C\,m}^{-3}$ for OPEM-H (Figure 4D), although DIC data have not been explicitly considered in the cost function.

While the trade-off solutions exhibit $NO_3^-$, $O_2$ and DIC inventories well within their respective uncertainty ranges, we find somewhat larger deviations for the predicted global mean net primary production (NPP, Figure 4E). For OPEM and OPEM-H

the trade-off solutions produce a, respectively, $30\%$ and $14\%$ higher NPP than the minimum-cost solutions. The net community production (NCP, here integrated over the depth range 0 to $980\,\mathrm{m}$) estimates in Figure 4F are better constrained than NPP for both configurations. The trade-off solution of OPEM corresponds to a global NCP of $1.043\,\mathrm{Tg\,C\,year^{-1}}$, which is close to the trade-off estimate of OPEM-H, where $\mathrm{NCP} = 1.039\,\mathrm{Tg\,C\,year^{-1}}$.

Figure 5 shows globally averaged concentrations of $O_2$ versus $NO_3^-$ of all ensemble members. The spread of the ensembles differs between the two tracers (by a factor of two for $O_2$ and by a factor of six for $NO_3^-$). Most solutions overestimate the global average $NO_3^-$ concentration obtained from the WOA 2013 (Garcia et al., 2013a, b) and underestimate $O_2$. Solutions where both tracers strongly underestimate the WOA 2013 data are penalised by the cost function (Figure 5). The minimum-cost and trade-off solutions of OPEM and OPEM-H are close to the WOA 2013 estimates. The respective optimal solutions have

slightly higher global mean $O_2$ concentrations than the WOA 2013 and are in good agreement with respect to $NO_3^-$. In spite of larger costs, the trade-off solutions of both OPEM and OPEM-H are closer to the WOA 2013 estimate than the minimum-cost solutions (Figure 5). The ensemble solutions are unevenly spread around the WOA 2013 data-based estimates. This highlights that our trade-off solutions could not have been identified had we only considered the ensemble means.

     Figures 6 and 7 show zonally averaged $NO_3^-$ and $O_2$ in simulations with the lowest and highest $NO_3^-$ and the trade-off

simulation in the OPEM configuration. The high-$NO_3^-$ simulation has similar $NO_3^-$ and $O_2$ patterns to the trade-off simulation, despite the very different mean $NO_3^-$ and $O_2$ concentrations. The patterns are different in the low-$NO_3^-$ simulation because of stronger deoxygenation and denitrification, which occur mostly in North Pacific deep water. The greater similarity of global mean $O_2$ than $NO_3^-$ reflects the influence of atmospheric $O_2$ but also indicates that $NO_3^-$ is more sensitive to changes in the physiology of the diazotrophs.

**3.2.2    How well can model parameters be constrained?**

Cost is conspicuously correlated only with $\nu_{\mathrm{det}}$, $Q^{\mathrm{N}}_{0,\,\mathrm{phy}}$, and $\phi_{\mathrm{dia}}$ (Figure 8). $O_2$ and $NO_3^-$ are sensitive to $\nu_{\mathrm{det}}$ and $Q^{\mathrm{N}}_{0,\,\mathrm{phy}}$ but not to $\phi_{\mathrm{dia}}$ (Figure 1), which indicates that $\phi_{\mathrm{dia}}$ becomes more important at lower-cost simulations. The minimum-cost and trade-off simulations in OPEM and OPEM-H are usually closer to each other when parameters show strong correlations with costs (Figure 8).

Figure 9 shows how different biomes contribute to the misfit and variance parts of the total cost. For simulations with high cost function values ($J > 10^{10}$), we find the variance term to be dominant in the deep ocean (below 550 m). Among the 17 biomes this is well expressed in NP.SPSS (North Pacific subpolar seasonally stratified), NP.STSS (North Pacific subtropical seasonally stratified), NP.STPS (North Pacific subtropical permanently stratified), Pac.EQU.E (Eastern Pacific equatorial), Pac.EQU.W (Western Pacific equatorial), and IND.STPS (Indian Ocean subtropical permanently stratified) biomes,

overwhelming contributions from all other parts of the cost function and all other biomes for the 100 simulations with the highest total costs. These high-cost simulations tend to have low $NO_3^-$ and $O_2$ concentrations (Figure 5). Low $NO_3^-$ concentrations are coupled to low $O_2$ because of intense denitrification in the oxygen deficient zones (ODZs). Accordingly, simulations with very low $NO_3^-$ inventories suffer from widespread ODZs, occupying much of the deep water in the northern and equatorial

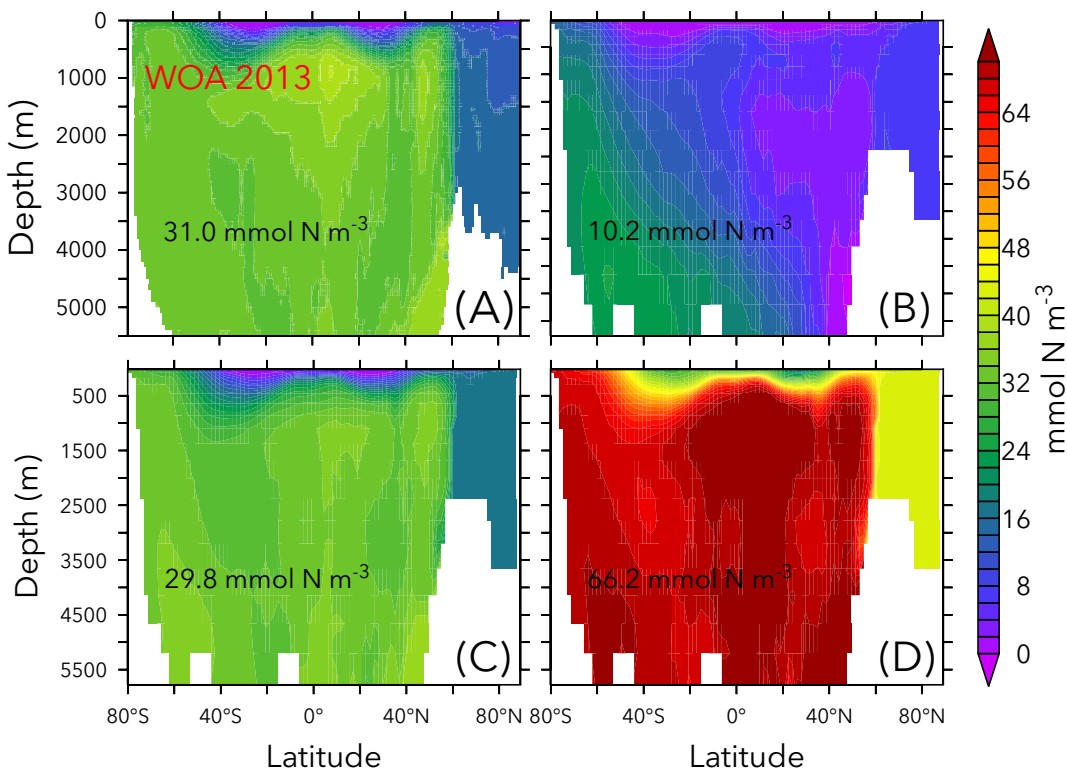

**Figure 6.** Zonally averaged $NO_3^-$ in the World Ocean Atlas 2013 (A), the simulations with the lowest and highest $NO_3^-$ inventory (B, D), and the trade-off simulation (C) in the OPEM configuration. Globally averaged $NO_3^-$ concentrations are shown in each panel. Simulations shown here are marked with solid black and open red triangles in Figure 5. Note that the outputs from OPEM and OPEM-H are very similar and only OPEM results are shown here.

Pacific as well as the Indian Ocean (Figure S1). This is the main reason for the high variance in the deep water of these biomes
(Figure 9).

## 4 Discussion

### 4.1 Parameter sensitivities

### 4.1.1 Remineralisation rate $\nu_{det}$ and phytoplankton subsistence nitrogen quota $Q_{0,\,phy}^N$

Remineralisation rate ($\nu_{det}$) and phytoplankton subsistence nitrogen quota ($Q_{0,\,phy}^N$) are the two parameters with the strongest
correlations for most tracers as well as particulate elemental stoichiometry. The importance of $\nu_{det}$ was expected, because it is an important driver of nutrient recycling in the surface ocean (Thomas, 2002; Anderson and Sarmiento, 1994; Eppley and

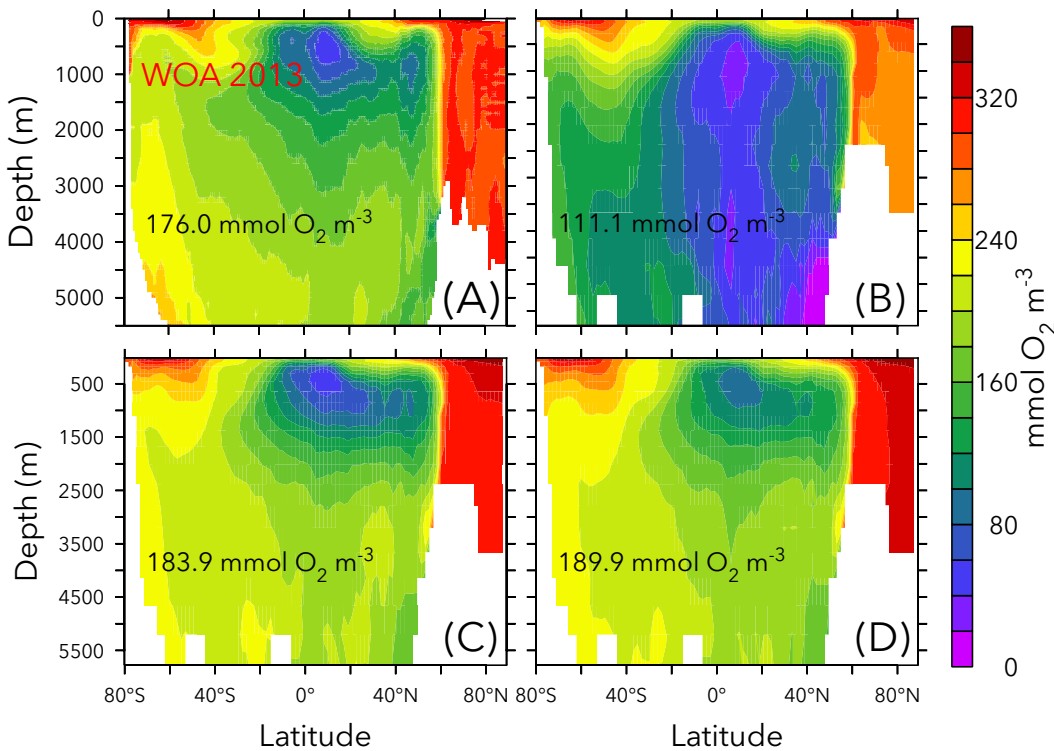

**Figure 7.** Same simulations as in Figure 6 but showing the results for $O_2$.

Peterson, 1979), which strongly affects NPP, NCP, Chl, DIC, DFe and $N_2$ fixation (Kriest et al., 2012). $\nu_{det}$ also determines the rate of $O_2$ consumption, hence also the $NO_3^-$ level, due to denitrification in ODZs (Cavan et al., 2017). The strong influence of $Q^N_{0,\,phy}$, however, was unexpected. The subsistence quota was first introduced by Droop (1968) in phytoplankton growth
models. While it has been applied in Earth System Models (Kwiatkowski et al., 2018; Wang et al., 2019), a sensitivity analysis similar to the present study has not been done before. A higher $Q^N_{0,\,phy}$ implies that more nitrogen is required for phytoplankton growth, but it also can be interpreted as a lessening of carbon fixation for a given nitrogen supply. Our results demonstrate a strong effect of $Q^N_{0,\,phy}$ on NPP, Chl, POC export (NCP, here integrated over the depth range $0$ to $980\,\mathrm{m}$) and consequently oxygen consumption and denitrification.

These results also put forward a new point of view on the relation between $NO_3^-$ inventory and carbon export. In classic biogeochemistry, a larger $NO_3^-$ inventory in the ocean stimulates primary production and POC export. This feedback is intuitive and easy to understand, as for a given C:N in phytoplankton, carbon is proportional to the nitrogen pool. This feedback is well recognized and has been widely applied in marine sciences, especially since it forms the foundation of one of the hypotheses explaining the lower atmospheric $pCO_2$ during the last glacial maximum (LGM) (McElroy, 1983; Falkowski, 1997). However,
our analysis of the model ensemble with different parameter combinations suggests another, very different point of view. $NO_3^-$

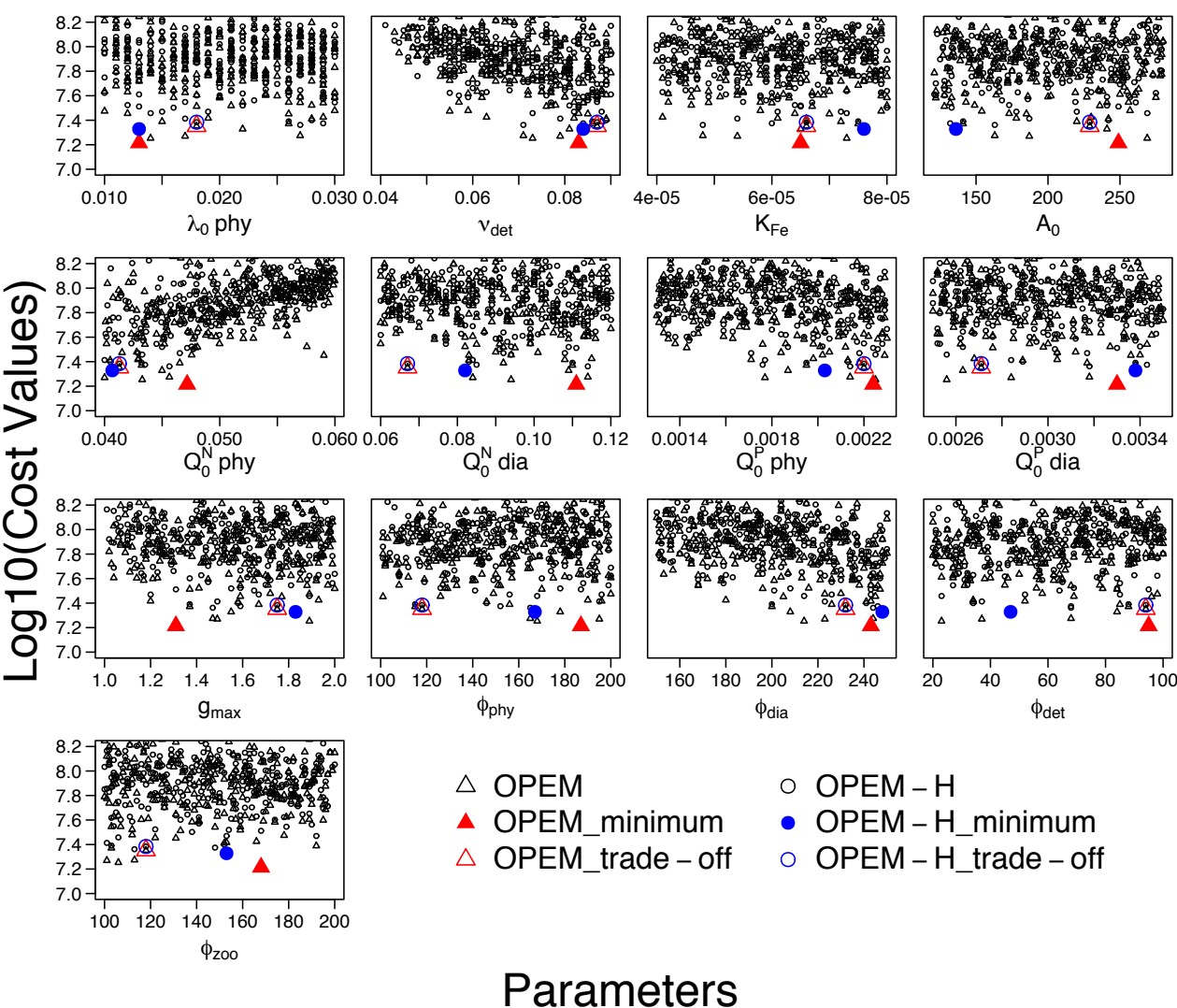

**Figure 8.** Lower parts (cost $< 10^{8.2}$) of cost-value distributions for the parameter ranges in Table 1. Solid red triangles and blue circles represent the minimum-cost simulations in OPEM and OPEM-H, respectively, and open red triangles and blue circles are the trade-off simulations. Note that the trade-off simulations share the same parameter combination but have slightly different cost-values.

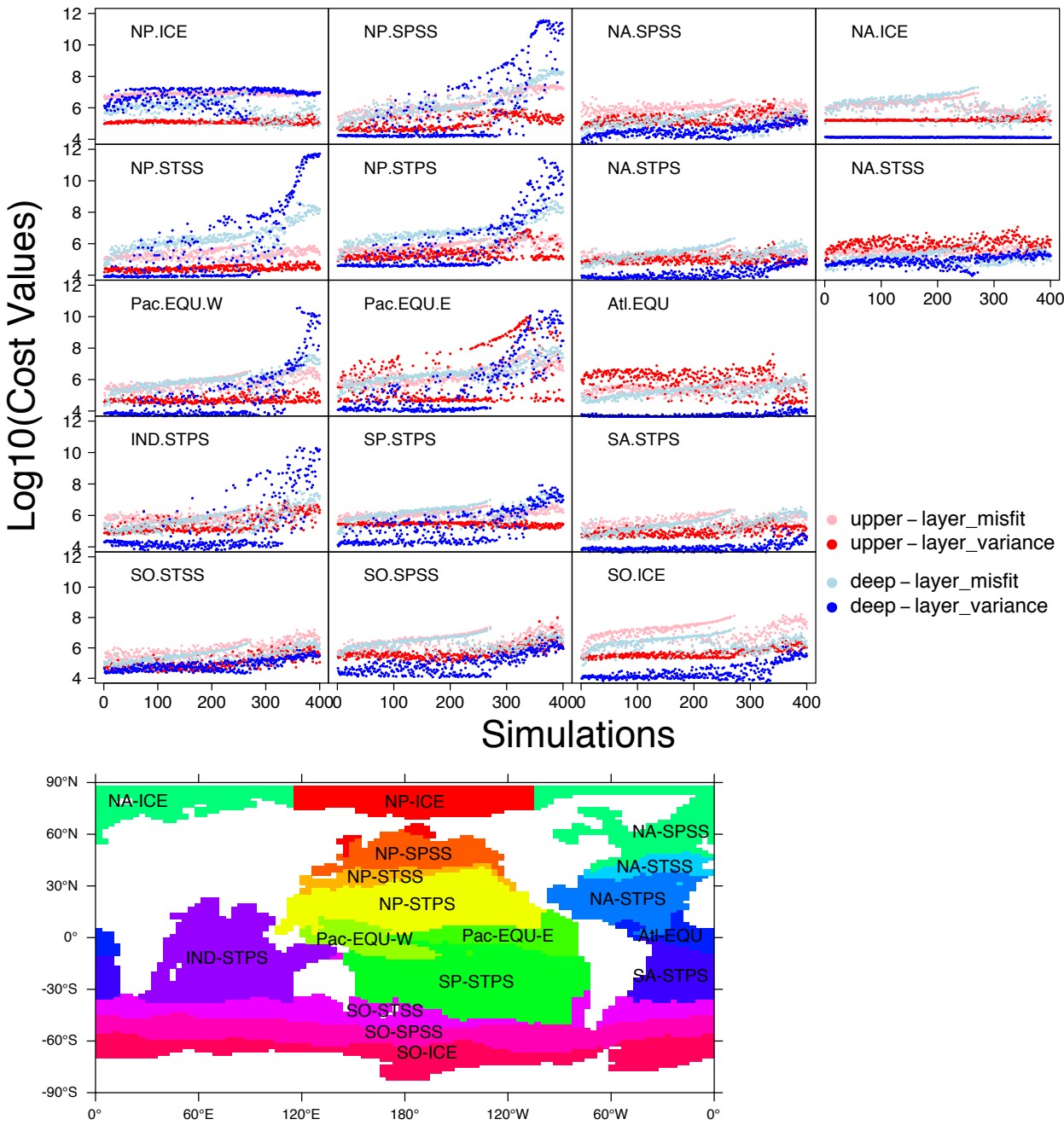

**Figure 9.** Top panels: Cost-value distributions in the 17 biomes in OPEM. The order of the simulations is based on the total cost from low to high in OPEM. Upper-layer and deep-layer in the legend represent upper ($0-550\,\mathrm{m}$) and lower (below $550\,\mathrm{m}$) components of the cost function (Eq. 5). Misfit and variance are calculated by the first and second parts of the cost function components (Eqs. 6 and 7), respectively. Bottom: Map of biome locations.

concentration is positively correlated with $Q_{0,\,\mathrm{phy}}^{\mathrm{N}}$, but negatively with NPP and POC export (NCP, Figure 1), which means that an increased $NO_3^-$ inventory can be related to a lower POC export if caused by a change in $Q_{0,\,\mathrm{phy}}^{\mathrm{N}}$. The dynamic C:N ratio in our model explains part of this negative correlation. When the $NO_3^-$ inventory increases due to an increase in $Q_{0,\,\mathrm{phy}}^{\mathrm{N}}$, the nitrogen demand in phytoplankton also increases, which yields a lower C:N ratio in phytoplankton, and hence changes in carbon fixation due to increases in $NO_3^-$ inventory remain relatively small. The increase in $Q_{0,\,\mathrm{phy}}^{\mathrm{N}}$ increases nitrogen in phytoplankton structure and decreases the C:N ratio in phytoplankton as well as detritus. The two effects together both lower POC production and raise the $NO_3^-$ inventory. Changes in $\nu_{\mathrm{det}}$ also contribute to the negative correlation between $NO_3^-$ and POC export (NCP) in our simulations: a more intense remineralisation in the surface ocean reduces POC export, and thus decreases oxygen consumption and denitrification, resulting in a larger nitrate inventory.

The strong impact of $Q_{0,\,\mathrm{phy}}^{\mathrm{N}}$ on the $NO_3^-$ inventory and globally averaged phytoplankton C:N causes a higher sensitivity of globally averaged C:N than C:P (Figure 3). A higher $Q_{0,\,\mathrm{phy}}^{\mathrm{N}}$ results in a higher $NO_3^-$ inventory and a lower phytoplankton C:N, both tending to lower particulate C:N and vice versa. On the other hand, C:P is not as sensitive because we have a constant $PO_4^{3-}$ inventory in the UVic model. Surface particulate matter C:N is less variable compared to C:P and N:P in field observations along regional gradients (Galbraith and Martiny, 2015; Geider and Roche, 2002; Martiny et al., 2013a; Sterner and Elser, 2002), which is an apparent contrast to our results, where the sensitivity of C:N to $Q_{0,\,\mathrm{phy}}^{\mathrm{N}}$ is the highest among the particulate elemental ratios. However, our sensitivities are with respect to parameter variations among many simulations, rather than spatial or temporal gradients in the one real ocean.

### 4.1.2 Zooplankton parameters

While in many global biogeochemical models zooplankton is described by non-mechanistic formulations, such as Holling-type functions (Holling and Buckingham, 1976), in this study we apply a more realistic zooplankton model (Pahlow and Prowe, 2010). Among the five zooplankton parameters, the maximum specific ingestion rate ($g_{\mathrm{max}}$) and the capture coefficients of phytoplankton ($\phi_{\mathrm{phy}}$) and diazotrophs ($\phi_{\mathrm{dia}}$) are the most important, whereas the preference for detritus ($\phi_{\mathrm{det}}$) is generally less important. Grazing on zooplankton itself ($\phi_{\mathrm{zoo}}$) counters the effect of $g_{\mathrm{max}}$ because it lowers zooplankton biomass and thus total ingestion. These parameters together dominate controls on $N_2$ fixation and Chl (Figure 1), and C, N and P of ordinary phytoplankton and diazotrophs (Figure 2). It is interesting that zooplankton parameters also exert some control on particulate N:P as well as the dissolved nutrient pools (Figure 3). This can be understood via their controls on $N_2$ fixation and the ensuing changes in N:P in the dissolved and particulate pools.

### 4.1.3 Other parameters and the OPEM-H configuration

Other parameters in the sensitivity analysis appear less important for the tracer distributions, but this does not necessarily mean that they are negligible. Specific mortality rate ($\lambda_{0,\,\mathrm{phy}}$) and the phytoplankton half-saturation constant for Fe ($k_{\mathrm{Fe,\,phy}}$) do contribute to some variations of most of the tracers (Figure 1), and particulate C:P is somewhat sensitive to potential nutrient affinity ($A_0$). Phytoplankton subsistence P quota ($Q_{0,\,\mathrm{phy}}^{\mathrm{P}}$) affects major tracers much less than phytoplankton subsistence N quota ($Q_{0,\,\mathrm{phy}}^{\mathrm{N}}$), but it is still important for particulate C:P and particulate N:P ratios, particularly at high latitudes and globally

(Figure 3). Diazotroph subsistence N and P quotas ($Q_{0,\,\mathrm{dia}}^{\mathrm{N}}$ and $Q_{0,\,\mathrm{dia}}^{\mathrm{P}}$) in general have much less influence on particulate stoichiometry than $Q_{0,\,\mathrm{phy}}^{\mathrm{N}}$ and $Q_{0,\,\mathrm{phy}}^{\mathrm{P}}$ because diazotrophs are much less abundant than ordinary phytoplankton. However, diazotroph biomass (carbon) itself is more sensitive to $Q_{0,\,\mathrm{dia}}^{\mathrm{N}}$ than $Q_{0,\,\mathrm{phy}}^{\mathrm{N}}$, which shows that the diazotroph subsistence quotas are still important for both their elemental stoichiometry and ability to compete with ordinary phytoplankton. While elemental stoichiometry has been suggested to be an important factor for determining the outcome of the competition between diazotrophs and non-diazotrophs, and consequently $N_2$ fixation (Deutsch and Weber, 2012; Weber and Deutsch, 2012), we find that $N_2$ fixation is no more sensitive to $Q_{0,\,\mathrm{dia}}^{\mathrm{N}}$ than to the remineralisation rate ($\nu_{\mathrm{det}}$), $Q_{0,\,\mathrm{phy}}^{\mathrm{N}}$, or zooplankton grazing parameters ($g_{\mathrm{max}}$, $\phi_{\mathrm{phy}}$, and $\phi_{\mathrm{dia}}$). Nevertheless, our analysis agrees with the argument that global $N_2$ fixation is mainly determined by rates of fixed-N loss (Weber and Deutsch, 2014), which in our model is largely affected by $\nu_{\mathrm{det}}$ and $Q_{0,\,\mathrm{phy}}^{\mathrm{N}}$.

In general, tracer sensitivities to parameters in OPEM-H configuration are similar to those in OPEM. $O_2$ and $NO_3^-$ levels are slightly less sensitive to the remineralisation rate, $Q_{0,\,\mathrm{phy}}^{\mathrm{N}}$, and $g_{\mathrm{max}}$ in OPEM-H because this configuration allows (facultative) diazotroph to grow in high-latitude cold waters, hence the overall biomass of diazotrophs is greater (Part I, Pahlow et al., 2020). This is also the reason why $Q_{0,\,\mathrm{dia}}^{\mathrm{N}}$ and $Q_{0,\,\mathrm{dia}}^{\mathrm{P}}$ exert a stronger effect on surface-particle elemental stoichiometry at high latitudes in OPEM-H (Figure 3).

Several studies have revealed that $N_2$ fixation occurs at high latitude regions (Sipler et al., 2017; Harding et al., 2018; Shiozaki et al., 2018; Mulholland et al., 2019), which supports a wider temperature range of $N_2$ fixation, similar to what we have in OPEM-H. In the trade-off simulation for OPEM-H we do find some $N_2$ fixation in the eastern North Pacific and the Arctic Ocean (Part I, Pahlow et al., 2020). The different temperature function for diazotrophy is also the reason for the differences in the sensitivities of particulate C:N:P to diazotroph subsistence quotas in high-latitude regions (Figure 3).

## 4.2 Model limitations

The strong correlation between $O_2$ and $NO_3^-$ (Figure 5) indicates that $O_2$ and denitrification are tightly coupled. Lack of benthic denitrification leaves water column denitrification as the only loss of $NO_3^-$ and $O_2$ becomes the primary factor controlling the $NO_3^-$ inventory. This implies that sensitivities of $NO_3^-$ to the model-parameters could be different when benthic denitrification is incorporated in our model. Also, this means that global $N_2$ fixation (same as global denitrification in our spun-up steady-state simulations) is underestimated, and since it occurs mostly at 40°S to 40°N (see Fig. 13 in Part I, Pahlow et al., 2020), particulate carbon to nitrogen (C:N) ratios could be overestimated due to a missing input of nitrogen to the surface ocean. This could explain the overestimated surface particulate C:N at low latitudes (see Table 3 and Figure 16 in Part I, Pahlow et al., 2020).

To evaluate how water-column denitrification affects our cost function, we arrange our simulations in the order of their cost values and plot the volume of oxygen deficient zones (ODZs) against cost for both the OPEM and OPEM-H configurations in Figure 10A to C. Several of our simulations, mostly among those with the 200 lowest cost values (Figure 10A), have a relatively small misfit in $O_2$ and $NO_3^-$ compared to the WOA 2013, and high $N_2$ fixation rates, comparable to those estimated in previous model studies (e.g., Somes et al., 2017; Wang et al., 2019). For these simulations, low $O_2$ is connected with high rates of water-column denitrification in the eastern equatorial Pacific Ocean (Pac.EQU.E), causing a depression of $NO_3^-$

concentration and a rather high variance in $NO_3^-$ concentration, both of which conflict with the observations. Hence cost in this biome is very high, especially in the upper $550\,m$ (Figure 9), where denitrification is strongest. On the other hand, although the volume of ODZs in the minium-cost simulations in OPEM and OPEM-H is greater than in the WOA 2013 (Figure 10C), they yield rather low $N_2$ fixation rates (38.8 and $35.1\,Tg\,N\,year^{-1}$ for OPEM and OPEM-H, respectively). ODZ volumes in the trade-off simulations are more than twice that in the WOA 2013 (Figure 10) and yield global $N_2$ fixation rates close to current estimates of water-column denitrification (about $70\,Tg\,N\,year^{-1}$, Somes et al., 2017; Wang et al., 2019). The mismatch between ODZ volume and $N_2$ fixation rate indicates that a refined description of water-column denitrification setting may be needed (Sauerland et al., 2019). While the physical component (ocean circulation) of the UVic model is also very important for the global distribution of oxygen and nitrate, our results suggest that, clearly, only by considering all major nitrogen sources and sinks, such as atmospheric deposition and benthic denitrification, a better representation of $N_2$ fixation and the global marine nitrogen cycle can be achieved.

## 4.3 Likelihood-based metric

### 4.3.1 Applicability of the cost function and usefulness of introducing variance information

The cost function introduced above is a metric that quantifies the discrepancy between objectively analyzed observational data and simulation results. Our cost function proves useful for exploring the 400 ensemble model solutions and identifies model solutions that reproduce deep ocean gradients in the $NO_3^-$:$PO_4^{3-}$ ratio better than a classic fixed-stoichiometry model (Part I, Pahlow et al., 2020). In addition, the optimal model solutions yield improved NCP rate estimates integrated over the top 100m (Part I, Pahlow et al., 2020). In particular, the trade-off solutions of OPEM and OPEM-H can resolve observed latitudinal patterns in dissolved and particulate C:N:P within the upper productive ocean layers (0–130 m, see Part I, Pahlow et al., 2020). The consideration of monthly mean $O_2$, $NO_3^-$, $PO_4^{3-}$ data for the upper $550\,m$ and surface Chl remote sensing data introduces important constraints on the representation of the relation between light and nutrient limitation, thereby also specifying the degrees of N and P limitation.

Even within the 5% of the simulations with the lowest costs, the estimates of global $N_2$ fixation rate vary considerably. The mean global estimates $\pm$ standard deviation in OPEM and OPEM-H are $(37 \pm 26)\,Tg\,N\,yr^{-1}$ and $(51 \pm 29)\,Tg\,N\,yr^{-1}$, respectively. We initially expected that the $NO_3^-$ and $PO_4^{3-}$ data in the cost function would effactually constrain $N_2$ fixation. This is clearly not the case and additional information has to be considered. One explanation may be that considerable $N_2$ fixation can occur during short periods and may also be confined to regions smaller than the biomes. Regional differences with respect to $N_2$ fixation remain unresolved if only biome-specific monthly mean $NO_3^-$ and $PO_4^{3-}$ data are considered for the upper layers in the cost function.

Also, the minimum-cost solution yields very low global $N_2$ fixation rates. Thus, for the identification of the trade-off solutions we had to consider prior information about global water column denitrification, whose rate is balanced by $N_2$ fixation according to our models. Incorporating $N_2$ fixation as a single global rate estimate into our Likelihood-based cost function as a single additional term would, without some difficult-to-define regulatization, become overwhelmed by the many tracer and variance

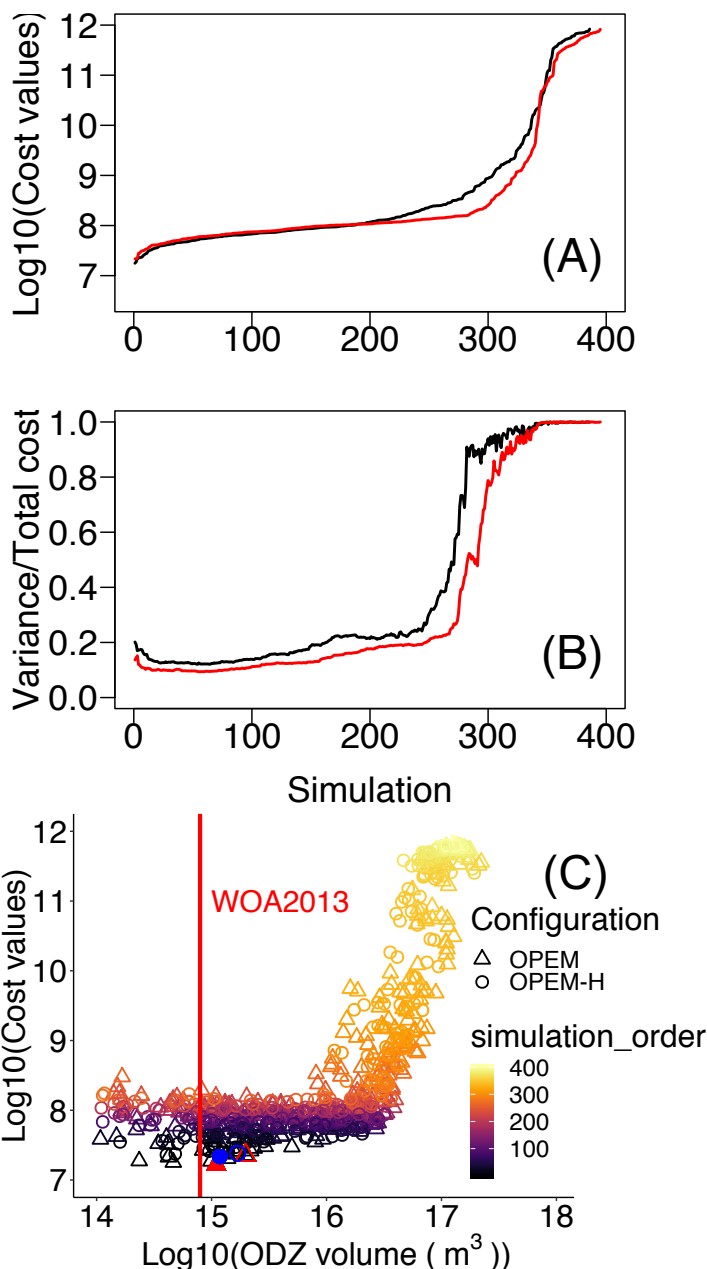

**Figure 10.** Cost values across all parameter sensitivity simulations ordered from low to high for the two model configurations. Cost values in both misfit and variance (A) and the contributions of variance (B). Black and red lines are for OPEM and OPEM-H, respectively. Total cost versus volume of ODZ (oxygen deficient zone $< 5\,\mathrm{mmol\,O\,m^{-3}}$) in the simulations (C), color represents the simulation order as shown in (A) and (B). The red vertical line indicates ODZ volume in the WOA 2013 ($7.45 \times 10^{14}\mathrm{m^3}$), the solid red triangle and blue circle represent the simulations with minimum cost in OPEM and OPEM-H, respectively, and open red triangle and blue circle are the trade-off simulations.

terms defined in Eqs. (6) and (7). Rather, the additional information is treated as a second objective, namely that global $N_2$ fixation should be greater than $60 \, \mathrm{Tg \, N \, yr^{-1}}$ (see above), which is similar to applying a multi-objective approach for model calibration (e.g., Sauerland et al., 2019), where a trade-off between two or more objectives (cost functions) is resolved. A refined cost function may incorporate monthly mean N:P ratios or N* values based on WOA 2013 data (e.g., for the upper 130 m) for clustered sub-regions of some biomes. Such addition to the cost function would require some careful preprocessing, e.g., cluster analysis of the spatial N:P or N* patterns, but may suffice to constrain simulated $N_2$ fixation rates.

A peculiarity of our cost function is that it complements the data-model misfit, i.e. the residuals of spatial mean $\log_{10}$-transformed values, with an additional term that resolves differences in spatial variances. How the neglect of this term affects the global mean tracer concentrations and flux estimates is depicted in Figures (S2 – S7) in the supplemental material. The cost function's variance term introduces a strong penalty to approximately $30\,\%$ of all ensemble model solutions. The highest cost-function values ($J > 10^9$) are associated with discrepancies in spatial variances that exceed the misfits in the $\log_{10}$-transformed tracer concentrations. For large parts of the ensemble solutions the variance term contributes between 15 and $20\,\%$ to the total costs. Interestingly, for those model solutions that yield low cost function values ($J < 4 \times 10^7$) the relative contribution rises again when the misfit in the $\log_{10}$-transformed tracer concentrations gradually decreases (Figure 10B).

### 4.3.2 Contributions of biomes

The 17 biomes derived by Fay and McKinley (2014) represent a scale similar to that addressed in global efforts to establish surface-ocean air-sea carbon-flux estimates (Wanninkhof et al., 2013; Rödenbeck et al., 2015). Accordingly, our cost function can be easily extended by incorporating air-sea $CO_2$ flux estimates in the future. Further improvements may be possible by introducing sub-regions in some biomes, e.g., for constraining $N_2$ fixation rate estimates, as discussed above.

For low cost function values the contribution of the variance term is generally small in most biomes for the deep layers (Figure 9), where variances of the $\log_{10}$-transformed tracer concentrations compare very well between the simulations and the WOA 2013. For high costs this term can become dominant, e.g., for some biomes in the North Pacific as well as the Indian Ocean. A remarkable exception is the North Pacific Arctic biome (NP-ICE), where the deep layer's variance term remains dominant for most of the ensemble solutions. This is somewhat different in the Arctic biome of the North Atlantic (NA-ICE) and the Southern Ocean (SO-ICE), where the variance term remains low throughout almost the entire ensemble. For SO-ICE the cost function is mainly affected by the misfit in $\log_{10}$-transformed tracer concentrations. The misfit is associated mainly with discrepancies between observed and simulated $NO_3^-$ within the SO-ICE biome. Interestingly, these misfits in both upper and deeper layers drop again after around the 280[th] simulation. Simulations with high $NO_3^-$ do not result in total cost values as high as in simulations with very low $NO_3^-$ (Figure 5), but they have larger misfits for $NO_3^-$ in SO-ICE. A similar behaviour can be seen in the other Southern Ocean biome (SO-SPSS) as well as in NA-ICE.

The upper layer's variance term contributes strongly for low costs in North Atlantic biomes. This is particularly striking for the Equatorial Atlantic biome (Atl-EQU). The main reason is water column denitrification that results in a high variance in $NO_3^-$. Likewise the Eastern Equatorial Pacific biome (Pac-EQU-E) reveals major model limitations in the upper layers. Overall, the unfolding of biome-specific contributions to the cost function clearly points to those regions where improving

model performance appears most worthwhile. Our present cost function may then be reapplied to quantify and highlight specific model improvements.

## 5 Conclusions

We demonstrate sensitivities of various tracers and processes to parameters in two configurations of a new optimality-based plankton-ecosystem model (OPEM) in the UVic-ESCM. While OPEM-H predicts a wider geographical range for $N_2$ fixation

(Part I, Pahlow et al., 2020) and shows some differences in the sensitivities of diazotroph C, N and P to parameters when compared to OPEM, the tracer sensitivity to model parameters is very similar in both configurations. The trade-off simulations in the OPEM and OPEM-H happen to have the same parameter set. Among our model simulations, varying model parameters within reasonable ranges results in variations in $O_2$ by a factor of two and in $NO_3^-$ concentration by a factor of six. The sensitivity analysis provides important information regarding the new models' behaviour. The $O_2$ inventory is mainly influenced

by the remineralisation rate ($\nu_{det}$) as well as phytoplankton subsistence nitrogen quota ($Q_{0,\,phy}^N$) and zooplankton maximum specific ingestion rate ($g_{max}$). Changes in $Q_{0,\,phy}^N$ strongly impact the $NO_3^-$ inventory, as well as the elemental stoichiometry of ordinary phytoplankton, diazotrophs and detritus. $Q_{0,\,phy}^N$ also affects $N_2$ fixation, Chl, DIC and iron levels. Furthermore, our sensitivity analysis resolves correlations between various biogeochemical tracers. For example, POC export is negatively correlated with the $NO_3^-$ inventory. We would like to point out that these changes in model behaviour are solely caused by

variations in parameters. Thus, the correlations between tracers and rates might not stand when tracer variations are caused by other factors. For example, an increase in the $NO_3^-$ inventory due to anthropogenic emissions may be accompanied by an increase in POC export (Fernández-Castro et al., 2016). Also, although we did evaluate sensitivities of particulate elemental stoichiometry at different latitudes, most tracer sensitivities and correlations should be considered valid only for global but not regional scales.

We introduce a new likelihood-based metric for model calibration. The metric appears capable of constraining globally averaged $O_2$, $NO_3^-$ and DIC concentrations as well as NCP. In particular, the minimum-cost and trade-off model solutions resolve observed latitudinal patterns in particulate C:N:P within the surface layers ($0 - 130\,\mathrm{m}$). However, the metric does not effectually constrain the models' global $N_2$ fixation rate estimates. Individual contributions of the biomes to the cost function provide details of how tracer distributions in each biome respond differently under different ecosystem settings.

The consideration of spatio-temporal variations in the stoichiometry of $NO_3^-$, $PO_4^{3-}$, and $O_2$ in our metric favours model solutions with low $N_2$ fixation rates that are solely balanced by low rates of water column denitrification. From our findings we conclude that an explicit consideration of benthic denitrification and atmospheric deposition seem critical for improving the representation of the complete global nitrogen cycle in our model.

*Code availability.* The University of Victoria Earth System Climate Model version 2.9 (Original Model) is available at http://www.climate.
uvic.ca/model/. The OPEM v1.0 code is available at http://dx.doi.org/10.3289/SW_1_2020. The instructions needed to reproduce the model
results described in this article are in the supplemental material.

*Author contributions.* Chia-Te Chien and Markus Pahlow performed the ensemble solutions and selected the reference simulations. Markus
Schartau set up the likelihood-based metric. All authors contributed to the manuscript text.

*Competing interests.* The authors declare that they have no conflict of interest.

*Acknowledgements.* Chia-Te Chien, Markus Pahlow and Markus Schartau were supported by the BMBF-funded project PalMod. Markus
Pahlow was supported by Deutsche Forschungsgemeinschaft (DFG) by the SFB754 (Sonderforschungsbereich 754 "Climate-Biogeochemistry
Interactions in the Tropical Ocean", www.sfb754.de) and as part of the Priority Programme 1704 (DynaTrait).

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
