# Peer review of "Optimality-Based Non-Redfield Plankton-Ecosystem Model (OPEM v1.0) in the UVic-ESCM 2.9. Part II: Sensitivity Analysis and Model Calibration Chia-Te Chien, Markus Pahlow, Markus Schartau, and Andreas Oschlies"

_Geoscientific Model Development, 2019_

## Referee Comment (RC1) · Sakina-Dorothée Ayata (Referee) · 12 Mar 2020

Referee comment on the manuscript "Optimality-Based Non-Redfield Plankton-Ecosystem Model (OPEM v1.0) in the UVic-ESCM 2.9. Part II: Sensitivity Analysis and Model Calibration", by Chia-Te Chien and colleagues.

**1) General comments**

This article presents the sensitivity analysis of an optimality-based plankton-ecosystem model (OPEM), implemented in the University of Victoria Earth-System Climate Model.

This model is described in an accompanying paper submitted to GMD (available as a discussion paper; Pahlow et al. 2019), while this article focuses on the evaluation of model performances. Given a set of biogeochemical parameter ranges, 400 simulations were performed using a Latin-Hypercube sampling method. This ensemble of model solutions was then used to select the best model parameterisations using a likelihood-based cost function taking into account both temporal and spatial variations of observed NO3-, PO43-, O2, and chlorophyll a concentrations at different depth levels. Two biogeochemical models with two different formulations of the temperature dependency for diazotroph's growth were considered (OPEM and OPEM-H) but led to the same choice of best parameter values. However, these best solutions led to low N2 fixation and denitrification at global scale, as these rates were poorly constrained by the data. Estimates of water-column denitrification were then used to identify the "best" model parameterisation within the ensemble of model solutions. A sensitivity analysis was also conducted for all biogeochemical tracers and all "optimized" parameters. The results revealed that the most important parameters for O2 concentration were the remineralisation rate, the subsistence N quota of ordinary phytoplankton, and zooplankton maximum specific ingestion rate, underlying the central role of phytoplankton physiology and elemental stoichiometry in global nitrogen cycle in the ocean. From their results and sensitivity analyses, the authors propose new hypotheses on the link between NO3- concentrations at global scale and phytoplankton physiology. As a perspective for their work, the authors suggest to explicitly include benthic denitrification and atmospheric deposition in future biogeochemical models to better represent the global nitrogen cycle.

The Introduction section is easy to follow and clearly presents the context and the aim of the study.

In the Materials  Methods section, the authors briefly present the models (the biogeochemical model is fully described in Pahlow et al. 2019). They detail the sensitivity analysis and the model calibration method. The later is based on the definition of a

likelihood-based cost function taking into account four different types of observations (monthly and/or annual climatologies of NO3-, PO43-, O2, and Chl a concentrations at various depth levels, averaged over 17 biogeochemical biomes). This original definition allows for taking into account spatial differences between biomes, as well as temporal differences at various depth levels, which is very good to fully constrain the spatial and temporal dynamics of the tracers.

The Results section then describes the ranges of global averages of major tracer concentrations (for both OPEM and OPEM-H), the sensitivity of biogeochemical tracer estimates (incl. phytoplankton biomass and stoichiometry) to model parameters, and a detailed description of the cost function values of the 400 simulations, especially for different global estimates of biogeochemical tracers.

The Discussion section focuses on 1) the sensitivity of key parameters (including remineralisation rate, phytoplankton subsistence nitrogen quota, and maximum specific ingestion rate of zooplankton), 2) the main model limitations (e.g., lack of benthic denitrification and atmospheric deposition may explain the high simulated volume of oxygen deficient zones compared to the observation), and 3) the insight provided by the use of their original cost function, especially on the usefulness of including variance information in this cost function and on the consideration of several biomes for its calculus.

The Conclusion section is clear. In this section, the main results are listed, the originality of the study is underlined, and the perspectives are indicated.

The text is well written and very clear. Most of the figures are properly described. Typo errors are extremely rare.

Given the very good scientific quality of the manuscript, I recommend minor revisions before publication. See below.

**Review items**

1) Does the paper address relevant scientific modelling questions within the scope of
GMD? YES Does the paper present a model, advances in modelling science, or a modelling protocol that is suitable for addressing relevant scientific questions within the scope of EGU? YES

2) Does the paper present novel concepts, ideas, tools, or data? YES

3) Does the paper represent a sufficiently substantial advance in modelling science? YES

4) Are the methods and assumptions valid and clearly outlined? YES

5) Are the results sufficient to support the interpretations and conclusions? YES

6) Is the description sufficiently complete and precise to allow their reproduction by fellow scientists (traceability of results)? In the case of model description papers, it should in theory be possible for an independent scientist to construct a model that, while not necessarily numerically identical, will produce scientifically equivalent results. Model development papers should be similarly reproducible. For MIP and benchmarking papers, it should be possible for the protocol to be precisely reproduced for an independent model. Descriptions of numerical advances should be precisely reproducible. YES

7) Do the authors give proper credit to related work and clearly indicate their own new/original contribution? YES

8) Does the title clearly reflect the contents of the paper? The model name and number should be included in papers that deal with only one model. YES

9) Does the abstract provide a concise and complete summary? YES

10) Is the overall presentation well structured and clear? YES

11) Is the language fluent and precise? YES

12) Are mathematical formulae, symbols, abbreviations, and units correctly defined and

used? YES

13) Should any parts of the paper (text, formulae, figures, tables) be clarified, reduced, combined, or eliminated? NO

14) Are the number and quality of references appropriate? YES

15) Is the amount and quality of supplementary material appropriate? For model description papers, authors are strongly encouraged to submit supplementary material containing the model code and a user manual. For development, technical, and benchmarking papers, the submission of code to perform calculations described in the text is strongly encouraged. YES

**2) Specific comments**

**Remarks on the Methods:**

-Equations 1 and 2: why don't you also test the sensitivity of the model to the parameters of these two equations? Please justify it in the text.

- Table 1: More clearly indicate in the legend of Table 1 that the identified "best" values for trade-off simulations were the same for the two model configurations OPEM and OPEM-H. It is indicated in the text later, but it has to be clearly mentioned here for the reader.

- Lines 102-107: I am not entirely convinced by the arguments given here to justify why the parallel setup is better than systematic calibration approaches. Indeed, one can imagine a systematic calibration where X values are systematically tested for the 13 parameters. In that case, I do not see how this would lead to individual model simulations that would depend on other/previous combinations of parameters, neither how it would prevent re-evaluation with different metrics. However, the first item would be true for a parameterisation based microgenetic algorithm for instance. The authors may need to rephrase this sentence to make it accurate (e.g., by replacing the term "systematic" by an other one).

[Figure]

**Remarks on the Results:**

- Table 2: Could you add a column with observation values, at least for the depth levels where the data are available (and also add rows of simulated values at these different depth levels), so that the reader can also estimate if the observed concentrations/fluxes fall into the range of simulated values? Otherwise, I have the feeling that this table could be removed. As it is, is in unclear, even from lines 170-175, which main result(s) the reader should keep in mind from this Table.

- Positive comment: Figure 1 nicely shows that the model outputs are highly sensitive to $\text{nu}_{d}\,et\,and\,Q_0^N\,phy\,(and\,to\,g_m\,ax\,and\,phy_p\,hy\,at\,the\,second\,order)$!

- Section 3.3.3 is quite long, but it presents a very detailed and interesting description and associated comments of the results presented in Figure 1. Keep it as it is.

- Figure 3: Justify in the methods the reason(s) of your choice of performing a regional splitting into latitudinal bands. This is missing in the article. Indeed, you mention later line 259: "sensitivities of dissolved N:P ratio to parameters in [...] three geographical settings (low, high latitudes and global)". It has to be mentioned (and justified) earlier.

- Line 250: "where diazotrophs are abundant in high latitudes": yet, this is not visible from your results. If this comes from Pahlow et al. 2019 please indicate it.

- Figure 2: Add a legend for black (OPEM) and grey (OPEM-H) as you did in Figure 1. Same comment for Figure 3.

- Line 285-288: It seems to me that these sentences should rather be included in the Methods section, not in the Results section.

- Figure 3: It may be nice to highlight the values that differ between OPEM and OPEM-H, for instance with rectangles (around the bars) and or stars (below or above the bars), so that it would clearly strike the eyes that the differences between the two configurations are obtained for the 60°S-70°S latitudinal band for C:N, C:P, and N:P. Also indicate in the legend the choices you made for the "different latitude bands".

- Figure 5 is not easy to read as it is because purple and black symbols look very similar. Smaller symbols may be used to help. Drawing horizontal and vertical lines to better underline the location of the WOA 2013 values (green square) may also be a good idea (although the figure is very well described lines 306-313).

- Lines 313-314: "Overall, we stress that the minimum-cost and trade-off solutions appear at the margin of the full spread of the ensembles, which could be interpreted as indicating a model deficiency.": I do not understand what you mean here. For me, it seems that they are in a patch of simulations with symbols in black, indicated $\log_{10}$ of cost values lower than 8, which seems OK. What are you referring to by the term "model deficiency"?

- Line 315: "Figures 6 and 7 show zonally averaged NO3 – and O2 in simulations with low and high NO3 – and the trade-off simulations": Would it be possible to delineate these simulations in Figure 5? Indeed, it is unclear if the concentrations presented Figures 6 and 7 come from one simulation only, or from several (how many?) simulations. When describing these two figures, also underline the fact that the outputs from OPEM and OPEM-H are very similar here. If this is indeed the trade-off simulation (as indicated in the legend), then the results should be the same and there is no need to show twice the same figures.

- Line 332: "because of intense denitrification in the ODZ" => the last (and first) time that you used the abbreviation ODZ was line 193. As it has not been used since, I recommend giving the full name here again and not just the abbreviation (as you do it later line 402). - Line 334: "widespread ODZs, occupying much of the deep water in the northern and equatorial Pacific as well as the Indian Ocean (Figure 6)" => Please indicate these areas clearly on the Figure 6, using arrows for instance.

- Figure 8: clearly mention in the legend that the two trade-off simulations for OPEM and OPEM-H are in fact the same, and use only one symbol for this trade-off simulation for figure clarity.

**Remarks on the Discussion:**

- The section 4.1.1. (especially the lines 348-362) provides new and very interesting hypotheses on the link between NO3 inventory at global scale and phytoplankton physiology. I appreciate this section.

- Line 404: "ODZ volumes in the trade-off simulations are more than twice that in the WOA 2013 (Figure 10)" => I do not see where it is visible on the Figure 10. I guess it could be inferred from Figure 10C from an expert eye, but I would rather give the precise value in the legend of Figure 10, with the corresponding vertical lines on Figure 10C, if you decide to keep the text as it is. Besides, this is the fist mention of Figure 10, that will be mentioned again line 439. I recommend clearly describing this figure here and later in the discussion, to fully explain and exploit it.

- Line 436-437: "A peculiarity of our cost function is that it complements the data-model misfit, i.e. the residuals of spatial mean log- transformed values, with an additional term that resolves differences in spatial variances" => Yes, indeed! I have particularly appreciated this.

- Line 439: "The cost function's variance term introduces a strong penalty to approximately 30

**Additional remarks:**

- I am wondering why keeping the quarter of the 400 simulations with the highest (worst) cost values in all the analyses, and not keeping only the 200 to 300 best ones?

**3) Technical corrections:**

**Minor comments and typos:**

- Lines 44-46: "Our new ecosystem model [...] offers new features and it improves the representation of some biogeochemical tracers on the global scale (see accompanying study, Pahlow et al. (2019)" => Which biogeochemical tracers? Give examples in brackets.

- Lines 48-49: "This model approach yields mass flux estimates with spatial and temporal variations in the elemental C:N:P stoichiometry of both inorganic nutrients and organic matter." => Add at the end of this sentence: "as observed in situ" and give some references to justify (e.g. Martiny, A.C., Vrugt, J.A., Primeau, F.W., Lomas, M.W., 2013. Regional variation in the particulate organic carbon to nitrogen ratio in the surface ocean. Global Biogeochem. Cycles 27, 1-9.)

- Line 79: "Our setup comprises ensembles of 400 simulations for each of two model configurations. The two model configurations differ in how temperature affects diazotrophy." => This could be replaced by "Our setup comprises ensembles of 400 simulations for each of the two model configurations that differ in how temperature affects diazotrophy."

- Line 102: "the parallel setup with different parameter combinations has a some advantages" => Remove "a".

- Line 103: Replace "Individual" by "individual".

- Legend of Figure 4: Replace "minmum-cost" by "minimum-cost".

- Line 337: a space is missing after the term "quota".

- Line 360: "our simulations: A more intense..." replace "A" by "a".

- Line 378: You may want to change "do contribute some variations to most of the tracers" by "do contribute to some variations of most of the tracers"

- Line 393: Figure 5 instead of Fig. 5 (for homogeneity).

- Line 421: " The mean global estimates $\pm 1$ standard deviation in OPEM and OPEM-H are..."=> You may want to replace "$\pm 1$" by "$\pm$".

- Line 496: "and" instead of "adn"

Please also note the supplement to this comment:

https://www.geosci-model-dev-discuss.net/gmd-2019-324/gmd-2019-324-RC1-supplement.pdf

---

## Referee Comment (RC2) · Anonymous Referee #2 · 30 Mar 2020

This paper aims to optimize and calibrate important parameters used in the lower trophic marine ecosystem component of UVic-ESM and is a companion paper to the model description paper (Pahlow et al., 2019). In this study, authors set up cost functions to minimize the misfit between model outputs and observations for nitrate, phosphate, dissolved oxygen, and surface chlorophyll-a. Of the 13 parameters they have chosen to calibrate, the subsistence N quota of phytoplankton and remineralization rate have the highest sensitivity.

Overall, the paper is nicely written and organized. Optimization schemes are well

described and the parameters are calibrated rigorously. However, I do have some important points that need to be clarified before I am ready to recommend publications of this paper in the GMD.

General Comments:

1. What is the "best" model choice? The authors state in line 7 – "For identifying the "best" model we therefore also consider... water-column denitrification". I was not ultimately clear after reading this paper, what the "best" model choice is. Is it OPEM/OPEM-H with the lowest overall total cost function or "trade-off" model which does not necessarily have the lowest cost function (7th best) but does best at representing N cycle? I may have missed this but if water-column denitrification and N2 fixation are indeed very important, why did you not include these in your cost function?

2. What is the selling point of this "optimized" flexible C:N:P model? Authors state that most NPZD models do not adequately describe the behavior of plankton physiology such as non-Redfieldian plankton stoichiometry. However, outside the UVic framework, there are quite a few ESMs in the market already with flexible C:N:P including those in CMIP5 (see Bopp et al., 2013) and CMIP6 (see Arora et al., 2019). There are also some studies that utilize Pahlow's phytoplankton model (Kwiatkowski et al., 2018, 2019). My question then is what is the selling point of this model over other existing models out there? Is it the computational efficiency and how useful is this model for studying climatic conditions such as the last glacial maximum or future projections (lines 39)? I think some discussions on model comparisons would be useful.

3. How sensitive is "sensitive"? Authors discuss the sensitivity of each parameter in Section 3.1 but one thing I find problematic is that all the graphs in Figure 1 – 3 have different y-scale increments. Since sensitivity is non-dimensional, they should ideally all have the same axis for a fair comparison since authors frequently say things like "Sensitivity of XXX is low" (e.g., line 196) or "No single parameter dominates sensitivity" (line 217). Although such rigorous statistical treatments may not be expected for this

kind of modeling work, I want some general clarifications on how authors interpreted whether something is very sensitive or not.

4. The highest sensitivity of C:N over C:P and N:P? Regarding the sensitivity, I was quite surprised looking at Figure 3 that C:N has much larger sensitivity compared to C:P and N:P. The current understanding in the scientific community is that C:N is more homeostatic compared to C(N):P for autotrophs, heterotrophs, and for detritus (Galbraith and Martiny, 2015; Geider and La Roche, 2002; Martiny et al., 2013; Sterner and Elser, 2002). Looking at the companion paper by Pahlow et al. (2019), steady-state C:N also seems to overestimate observation (Table 3 and Figure 7). I think this is an important point to address given that C:N (and therefore QoN) affects all aspects of the model output and that the whole point of this model is incorporating flexible C:N:P.

Specific comments:

Equations: Diazotrophy rate increases indefinitely with temperature with this formulation. But the growth rate of diazotrophs should hit the limit at some optimal value (e,g., 28 degrees Celsius for Trichodesmium; Breitbarth, E., A. Oschlies, and J. LaRoche (2007), Physiological constraints on the global distribution of Trichodesmium-effect of temperature on diazotrophy, Biogeosciences (BG), 4(1), 53–61). What is the justification of this temperature formulation? I feel like Eppley (1972) is not quite up to date.

Line 85: The temperature dependence of nitrogenase activity in the terrestrial system was used. Are there not any data from marine ecosystem literature?

Table 1: How are the "Range" chosen for these parameters?

Table 2: Maybe it would be nice to have some "target" values for comparison from WOA 2013 or other datasets.

Line 202: What are the sinks for DFe?

Figure 2: Phytoplankton (1st column) and diazotrophs (2nd column) have different y-axis range. For a fair comparison, they should have the same y range (at least for the

same given row).

Line 246: "their biomass is higher". What is "biomass"? Is it C quota or C+N+P or Chl? I do not see "biomass" in Figure 2.

Line 254-257: The logical behind explaining C:P pattern is not clear. Why does NO3:PO4 supply stoichiometry only affect low latitudes? Why that fact P-limitation is not present in S. Ocean explain the negative correlation between C:P and QoN?

L282: The description of "trade-off solutions". I went to Pahlow et al. (2019) but I could not easily locate where the discussion is. Could you direct me specifically to where it is?

Figures 6 and 7: What does "low nitrate" and "high nitrate" mean? I may have missed it but are they different model configurations or are they taken from different oceanographic regions?

Also Figures 6 and 7: It would be nice to have a zonal average from WOA 2013 for comparison.

Line 381: N:P of diazotrophs is critically important for determining the outcome of competition between diazotrophs and non-diazotrophs so it should be discussed in more depths here (e.g., Weber and Deutsch, 2012).

Line 407: I think authors should also mention the fact that physical component/ocean circulation is very important for the global distribution of oxygen and nitrate.

Reference:

Arora, V., Katavouta, A., Williams, R., Jones, C., Brovkin, V., Friedlingstein, P., Schwinger, J., Bopp, L., Boucher, O., Cadule, P., Chamberlain, M., Christian, J., Delire, C., Fisher, R., Hajima, T., Ilyina, T., Joetzjer, E., Kawamiya, M., Koven, C., Krasting, J., Law, R., Lawrence, D., Lenton, A., Lindsay, K., Pongratz, J., Raddatz, T., Séférian, R., Tachiiri, K., Tjiputra, J., Wiltshire, A., Wu, T. and Ziehn, T.: Carbon-concentration and

carbon-climate feedbacks in CMIP6 models, and their comparison to CMIP5 models, Biogeosciences Discuss., 1–124, doi:10.5194/bg-2019-473, 2019.

Bopp, L., Resplandy, L., Orr, J. C., Doney, S. C., Dunne, J. P., Gehlen, M., Halloran, P., Heinze, C., Ilyina, T., Séférian, R., Tjiputra, J. and Vichi, M.: Multiple stressors of ocean ecosystems in the 21st century: projections with CMIP5 models, Biogeosciences, 10(10), 6225–6245, doi:10.5194/bg-10-6225-2013, 2013.

Galbraith, E. D. and Martiny, A. C.: A simple nutrient-dependence mechanism for predicting the stoichiometry of marine ecosystems, Proc. Natl. Acad. Sci., 112(27), 8199–8204, doi:10.1073/pnas.1423917112, 2015.

Geider, R. and La Roche, J.: Redfield revisited: variability of C:N:P in marine microalgae and its biochemical basis, Eur. J. Phycol., 37(1), 1–17, doi:10.1017/S0967026201003456, 2002.

Kwiatkowski, L., Aumont, O., Bopp, L. and Ciais, P.: The Impact of Variable Phytoplankton Stoichiometry on Projections of Primary Production, Food Quality, and Carbon Uptake in the Global Ocean, Global Biogeochem. Cycles, 32(4), 516–528, doi:10.1002/2017GB005799, 2018.

Kwiatkowski, L., Aumont, O. and Bopp, L.: Consistent trophic amplification of marine biomass declines under climate change, Glob. Chang. Biol., 25(1), 218–229, doi:10.1111/gcb.14468, 2019.

Martiny, A. C., Pham, C. T. A., Primeau, F. W., Vrugt, J. A., Moore, J. K., Levin, S. A. and Lomas, M. W.: Strong latitudinal patterns in the elemental ratios of marine plankton and organic matter, Nat. Geosci., 6(4), 279–283, doi:10.1038/ngeo1757, 2013.

Sterner, R. W. and Elser, J. J.: Ecological stoichiometry: the biology of elements from molecules to the biosphere, Princeton University Press, Princeton, NJ., 2002.

Weber, T. S. and Deutsch, C. A.: Oceanic nitrogen reservoir regulated by plankton diversity and ocean circulation., Nature, 489(7416), 419–22, doi:10.1038/nature11357,

2012.

---

## Author Comment (AC1) · 26 May 2020

Thank you to two referees for your comments. They have been very helpful in revising the manuscript. Please see our detailed responses and the revised manuscript, which we will submit after finalising this discussion process.

---

## Author Response (AR1)

**Responses to the referees and changes to the manuscript**

We want to thank the two referees for their helpful and constructive reviews, which have greatly improved the manuscript. Below please find our responses to all of your points. The track-changes (latexdiff) version of the manuscript follows at the end of this pdf.

Dear Sakina-Dorothée,

We like to thank your for the effort and the time spent for a careful and positive review. Your comments are much appreciated, as they are constructive and helpful. We listed our responses to all of your points below and hope the manuscript is now satisfactory.

**Remarks on the Methods**

- **Equations 1 and 2:** *why don't you also test the sensitivity of the model to the parameters of these two equations? Please justify it in the text.*
- **Reply:** These two temperature configurations are empirical functions, directly simulating expected or observed temperature dependencies of  $N_2$  fixation. One of the main goals of this study is to compare a more realisitic temperature dependency (OPEM-H) to the default function used in the UVic (OPEM). Thus, our study compares two different model configurations. The structural difference becomes ultimately fixed by the temperature function employed, which includes the values assigned to the respective parameters. In addition, our focus is on variations in physiological parameters, but the parameters of the two different temperature equations have no clear physiological meaning. They determine (define) the two model configurations exclusively. We explain this now in the text on p. 4, lines 94–103: "Both of these equations are empirical functions directly simulating expected or observed temperature dependencies of  $N_2$  fixation. We consider Eq. (2) more realistic and hence analyse its effect on model behaviour. However, since the parameters in these two equations have no clearly identifiable physiological meaning, we consider a sensitivity analysis of the parameters in Eqs. (1) and (2) beyond the scope of the present study. Note that some models do not enforce any temperature limitation on nitrogen fixation (e.g., Dunne et al., 2012; Ilyina et al., 2013; Jickells et al., 2017). In the present ocean, waters colder than about 15 °C are generally replete with fixed inorganic nitrogen. For existing parameterisations of  $N_2$  fixation, which are functions of the nitrate deficit with respect to phosphate, there has been little indication of substantial impacts of the formulation of temperature control at low temperatures on the distribution of nitrogen fixation (Somes and Oschlies, 2015; Landolfi et al., 2017). Such differences in formulation may, however, gain importance in environmental conditions different from today's.".
- **Table 1:** More clearly indicate in the legend of Table 1 that the identified "best" values for trade-off simulations were the same for the two model configurations OPEM and OPEM-H. It is indicated in the text later, but it has to be clearly mentioned here for the reader.

**Reply:** We have added "Note that the trade-off simulations share the same parameter combination." to the table caption.

- Lines 102–107: I am not entirely convinced by the arguments given here to justify why the parallel setup is better than systematic calibration approaches. Indeed, one can imagine a systematic calibration where X values are systematically tested for the 13 parameters. In that case, I do not see how this would lead to individual model simulations that would depend on other/previous combinations of parameters, neither how it would prevent re-evaluation with different metrics. However, the first item would be true for a parameterisation based microgenetic algorithm for instance. The authors may need to rephrase this sentence to make it accurate (e.g., by replacing the term "systematic" by an other one).
- **Reply:** We agree, by "systematic" we originally thought of a typical path-dependent minimization (or maximization) in the parameter-cost function manifold, which is an iterative process. We replaced "systematic" with *"iterative*", now on p. 4, line 113.

**Remarks on the Results**

- **Table 2:** Could you add a column with observation values, at least for the depth levels where the data are available (and also add rows of simulated values at these different depth levels), so that the reader can also estimate if the observed concentrations/fluxes fall into the range of simulated values? Otherwise, I have the feeling that this table could be removed. As it is, is in unclear, even from lines 170–175, which main result(s) the reader should keep in mind from this Table.
- **Reply:** We added one column with observational estimates, based on global averages of either available observations or of data-based (data-driven) model results for those tracers/rates whose measurements are sparse on the global scale. For better comparison we adjusted the depth ranges accordingly. Now Chl concentration is the average of the top 50 m, and NCP is calculated from 0 to 100 m. We removed POC export and show only NCP, since at steady state it is equivalent to POC export flux.
- **Positive comment:** Figure 1 nicely shows that the model outputs are highly sensitive to  $\nu_{det}$  and  $Q_{0, phy}^N$  (and to  $g_{max}$  and  $\phi_{phy}$  at the second order)!
- **Section 3.3.3** *is quite long, but it presents a very detailed and interesting description and associated comments of the results presented in Figure 1. Keep it as it is.*
- Reply: We appreciate your positive comments.
- **Figure 3:** Justify in the methods the reason(s) of your choice of performing a regional splitting into latitudinal bands. This is missing in the article. Indeed, you mention later line 259: "sensitivities of dissolved N:P ratio to parameters in [...] three geographical settings (low, high latitudes and global)". It has to be mentioned (and justified) earlier.
- **Reply:** We added the justification in the Methods section, now on p. 5, lines 127–129: "We also evaluate the sensitivities of surface particulate elemental ratios (C:N, C:P and N:P), as well as nitrate to phosphate ratios for different latitude bands (40°S to 40°N, 60°S to 70°S, and globally). This is because dissolved and particulate elemental ratios in general show very different behaviour between lower and higher latitudes (Martiny et al., 2013a)."
- **Line 250:** "where diazotrophs are abundant in high latitudes": yet, this is not visible from your results. If this comes from Pahlow et al. 2019 please indicate it.
- **Reply:** We have added "(see Fig. 15 in Pahlow et al., 2020)" on p. 11, lines 271–272.
- Figure 2: Add a legend for black (OPEM) and grey (OPEM-H) as you did in Figure 1. Same comment for Figure 3.
- Reply: Done
- Line 285–288: It seems to me that these sentences should rather be included in the Methods section, not in the Results section.

**Reply:** We moved this part to the Methods section (now on p. 7, lines 185–189).

- **Figure 3:** It may be nice to highlight the values that differ between OPEM and OPEM-H, for instance with rectangles (around the bars) and or stars (below or above the bars), so that it would clearly strike the eyes that the differences between the two configurations are obtained for the 60°S-70°S latitudinal band for C:N, C:P, and N:P. Also indicate in the legend the choices you made for the "different latitude bands".
- **Reply:** We follow your suggestion and have added asterisks below or above the bars where sensitivities are very different between OPEM and OPEM-H. We also indicate our choices of our latitude bands in the legend.

- **Figure 5** *is not easy to read as it is because purple and black symbols look very similar. Smaller symbols may be used to help.* Drawing horizontal and vertical lines to better underline the location of the WOA 2013 values (green square) may also *be a good idea (although the figure is very well described lines 306–313).*
- **Reply:** We shrank size of the symbols, and changed the color key, which appears to make it easier to read. In additional to drawing lines to indicate the location of the WOA 2013, we added its nitrate and oxygen values in the legend.
- Lines 313–314: "Overall, we stress that the minimum-cost and trade-off solutions appear at the margin of the full spread of the ensembles, which could be interpreted as indicating a model deficiency.": I do not understand what you mean here. For me, it seems that they are in a patch of simulations with symbols in black, indicated log10 of cost values lower than 8, which seems OK. What are you referring to by the term "model deficiency"?
- **Reply:** Ideally, we would obtain the ensemble solutions evenly spread around the WOA 2013 data-based estimate. That the cost values close to this point are low is a necessary consequence of the definition of the cost function. Our results clearly show, however, that the mean of the ensemble solutions does not correspond to the observational estimate. Given all the non-linearities, this is not really surprising. Referring to this as a "model deficiency" might be an overly critical statement. We removed the term "model deficiency" and rephrased our explanation on p. 17, lines 332–333: *"The ensemble solutions are unevenly spread around the WOA2013 databased estimates. This highlights that our trade-off solutions could not have been identified had we only considered the ensemble means."*
- **Line 315:** "Figures 6 and 7 show zonally averaged  $NO_3^-$  and  $O_2$  in simulations with low and high  $NO_3^-$  and the trade-off simulations": Would it be possible to delineate these simulations in Figure 5? Indeed, it is unclear if the concentrations presented Figures 6 and 7 come from one simulation only, or from several (how many?) simulations. When describing these two figures, also underline the fact that the outputs from OPEM and OPEM-H are very similar here. If this is indeed the trade-off simulation (as indicated in the legend), then the results should be the same and there is no need to show twice the same figures.
- **Reply:** The panels in Figures 6 and 7 are zonally averaged  $NO_3^-$  and  $O_2$  from three simulations that result from the three parameter sets that generated the lowest and highest  $NO_3^-$  inventories, and the trade-off simulation in the OPEM configuration. We have revised the sentence to read *"Figures 6 and 7 show zonally averaged NO\_3^- and O\_2 in simulations with the lowest and highest NO\_3^- and the trade-off simulation in the OPEM configuration." on p. 17, lines 334–335. We indicate the low and high NO\_3^- simulations with solid black symbols in Figure 5, while the trade-off simulations are already highlighted. We now underline the fact that the results from OPEM and OPEM-H are very similar and removed the panels for OPEM-H from these figures. We have also added zonal averages from the WOA 2013 for comparison.*
- Line 332: "because of intense denitrification in the ODZ" => the last (and first) time that you used the abbreviation ODZ was line 193. As it has not been used since, I recommend giving the full name here again and not just the abbreviation (as you do it later line 402).
- **Reply:** We have added the full name "oxygen deficient zones", now on p. 17, line 352.
- **Line 334:** "widespread ODZs, occupying much of the deep water in the northern and equatorial Pacific as well as the Indian Ocean (Figure 6)" => Please indicate these areas clearly on the Figure 6, using arrows for instance.
- **Reply:** We have added in the supplement a global 2D map (Figure S1) showing oxygen concentrations in the deep water (1240 to 5490 m) for a simulation with very low globally averaged oxygen. This simulation is the same as the low oxygen OPEM simulation shown in Figure 6.

- **Figure 8:** *clearly mention in the legend that the two trade-off simulations for OPEM and OPEM-H are in fact the same, and use only one symbol for this trade-off simulation for figure clarity.*
- **Reply:** The two trade-off do have same parameter combination, but their costs are slightly different, so we need to keep both symbols. We have added *"Note that trade-off simulations share same parameter combination but have slightly different cost values."* in the caption.

**Remarks on the Discussion**

**The section 4.1.1.:** (especially the lines 348–362) provides new and very interesting hypotheses on the link between NO3 *inventory at global scale and phytoplankton physiology. I appreciate this section.*

**Reply:** We appreciate this comment.

- Line 404: "ODZ volumes in the trade-off simulations are more than twice that in the WOA 2013 (Figure 10)" => I do not see where it is visible on the Figure 10. I guess it could be inferred from Figure 10C from an expert eye, but I would rather give the precise value in the legend of Figure 10, with the corresponding vertical lines on Figure 10C, if you decide to keep the text as it is. Besides, this is the fist mention of Figure 10, that will be mentioned again line 439. I recommend clearly describing this figure here and later in the discussion, to fully explain and exploit it.
- **Reply:** We have added ODZ volume for the WOA2013 (7.45 × 1014m3) in the caption and also added a description of this figure earlier in this section on p. 23, lines 437–439: "*To evaluate how water-column denitrification affects the cost value of our simulations, we arrange our simulations in the order of their cost values and plot the volume of oxygen deficient zones* (*ODZs*) *against cost values for both the OPEM and OPEM-H configurations in Figure 10A to C.*". On line 439 we misplaced an extra (Figure10) here. This sentence is referring to Figure (S1 -S6) (now Figure (S2-S7)) in the previous sentence. We have removed the misplaced (Figure 10) now.
- **Line 436–437:** *"A peculiarity of our cost function is that it complements the data-model misfit, i.e. the residuals of spatial mean log-transformed values, with an additional term that resolves differences in spatial variances"* => Yes, indeed! I have particularly appreciated this.
- **Reply:** We are happy that this is perceived in a positive way.
- **Line 439:** *"The cost function's variance term introduces a strong penalty to approximately 30 % of all ensemble model solutions (Figure 10)." => As mentioned above, Figure 10 lacks a clear description. I do not see what in Figure 10 supports this, but I am sure the authors could give more explanation for helping the reader through this.*
- **Reply:** We apologize, because we misplaced an extra (Figure10) here. This sentence is actually referring to Figure (S1–S6) (now Figure (S2–S7)) in the previous sentence. We have removed the misplaced (Figure 10) now.

**Additional remarks**

- *I am wondering why keeping the quarter of the 400 simulations with the highest (worst) cost values in all the analyses, and not keeping only the 200 to 300 best ones?*
- **Reply:** This is a fair and meaningful comment, because it reflects an aspect we also discussed internally. We concluded that our analyses should involve a global (total / full) sensitivity analysis (sensitivities to all variations in the full parameter-cost function manifold), rather than local sensitivity analyses in the vicinity around the

trade-off solutions. This way we think we reveal information about the full model behaviour. From our internal discussion we learned that another difficulty would be to justify a threshold limit around the vicinity, which becomes even more subjective in our situation where the minima ("best") do not exactly match the trade-off solutions. There is no justification for why we could use only the best 200, 237, 299, or 300 solutions for our analysis. We have decided instead to add an explanation on p. 5, lines 129–130: "We keep all 400 simulations because we want to obtain the sensitivity information for the full parameter ranges.".

**Technical corrections**

Minor comments and typos:

- **Lines 44–46:** "Our new ecosystem model [...] offers new features and it improves the representation of some biogeochemical tracers on the global scale (see accompanying study, Pahlow et al., 2019)" => Which biogeochemical tracers? Give examples in brackets.
- **Reply:** We provide examples here, so now the sentence reads "Our new ecosystem model [...] offers new features and it improves the representation of some biogeochemical tracers on the global scale (e.g., net community production (NCP) and particulate C:N:P in the surface water, see Part I, Pahlow et al., 2020)." on p. 2, lines 45–47.
- Lines 48–49: "This model approach yields mass flux estimates with spatial and temporal variations in the elemental C:N:P stoichiometry of both inorganic nutrients and organic matter." => Add at the end of this sentence: "as observed in situ" and give some references to justify (e.g. Martiny, A.C., Vrugt, J.A., Primeau, F.W., Lomas, M.W., 2013. Regional variation in the particulate organic carbon to nitrogen ratio in the surface ocean. Global Biogeochem. Cycles 27, 1–9.)

**Reply:** We have added "as observed in situ" and cite Martiny et al. (2013) and Loh et al. (2000) now on p. 2, line 51.

**Line 79:** "Our setup comprises ensembles of 400 simulations for each of two model configurations. The two model configurations differ in how temperature affects diazotro- phy." => This could be replaced by "Our setup comprises ensembles of 400 simulations for each of the two model configurations that differ in how temperature affects diazotrophy."

**Reply:** Done, now the sentence is on p. 3, lines 85–86.

**Line 102:** *"the parallel setup with different parameter combinations has a some advantages" => Remove "a".*

**Reply:** Done, now the sentence is on p. 4, line 113.

Line 103: Replace "Individual" by "individual".

**Reply:** Done, now it is on p. 4, line 114.

Legend of Figure 4: Replace "minmum-cost" by "minimum-cost".

Reply: Done.

Line 337: a space is missing after the term "quota".

**Reply:** Done, now the space is on p. 18, line 358.

Line 360: "our simulations: A more intense..." replace "A" by "a".

**Reply:** Done, now it is on p. 22, line 383.

**Line 378:** You may want to change "do contribute some variations to most of the tracers" by "do contribute to some variations of most of the tracers"

**Reply:** Done, now the sentence is on p. 22, lines 405–406.

**Line 393:** *Figure 5 instead of Fig. 5 (for homogeneity).*

**Reply:** Done, now it is on p. 23, line 429.

**Line 421:** "The mean global estimates  $\pm 1$  standard deviation in OPEM and OPEM-H are..."=> You may want to replace " $\pm 1$ " by " $\pm$ ".

**Reply:** Done, now it is on p. 24, line 466.

Line 496: "and" instead of "adn"

**Reply:** Done, now it is on p. 28, line 542.

We thank the referee for the constructive review. The comments and questions are useful, which helped us to introduce changes and improve our manuscript. We listed our responses to all of your points below and hope the manuscript is now satisfactory.

**General comments**

This paper aims to optimize and calibrate important parameters used in the lower trophic marine ecosystem component of UVic-ESM and is a companion paper to the model description paper (Pahlow et al., 2019). In this study, authors set up cost functions to minimize the misfit between model outputs and observations for nitrate, phosphate, dissolved oxygen, and surface chlorophyll-a. Of the 13 parameters they have chosen to calibrate, the subsistence N quota of phytoplankton and remineralization rate have the highest sensitivity. Overall, the paper is nicely written and organized. Optimization schemes are well described and the parameters are calibrated rigorously. However, I do have some important points that need to be clarified before I am ready to recommend publications of this paper in the GMD.

- 1. What is the "best" model choice? The authors state in line 7 "For identifying the "best" model we therefore also consider ... water-column denitrification". I was not ultimately clear after reading this paper, what the "best" model choice is. Is it OPEM/OPEM-H with the lowest overall total cost function or "trade-off" model which does not necessarily have the lowest cost function (7th best) but does best at representing N cycle? I may have missed this but if water-column denitrification and N2 fixation are indeed very important, why did you not include these in your cost function? Reply: The "best" model solutions in line 7 refer to the trade-off simulations in each of the OPEM and OPEM-H configurations. We changed "best model solutions" on p. 1, line 8 to "reference parameter sets" to avoid this confusion. We also modified p. 1, lines 6–8 to: "The simulations closest to the data with respect to our metric exhibit very low rates of global  $N_2$  fixation and denitrification, indicating that in order to achieve rates consistent with independent estimates, additional contraints have to be applied in the calibration process. For identifying the reference *parameter sets...*" During our analysis we had considered the implementation of observed  $N_2$  fixation rates to our cost function, as suggested by the referee. But we quickly learned that this is not straightforward, mainly because of the scarcity of observed rates on the global scale, which introduces a large imbalance (between the many terms for each of  $NO_3^-$ ,  $O_2$ , and DIC, and the one term for global  $N_2$  fixation) in our cost function. The spatial and temporal coverage of these data is very different from the monthly resolved tracer concentrations we consider for our cost function. Such an imbalance requires the introduction of some regularization, which would make the cost function less objective than it is now. Instead, we interpreted the identified model solutions in terms of a multi-objective optimisation. In this manner, the consideration of global  $N_2$  fixation rates is treated as a second objective, in addition to our cost function being the first objective. This is the reason why we refer to these model results as trade-off solutions. We address the problem now more explicitly on pp. 24–26, lines 474–478: "Incorporating  $N_2$  fixation as a single global rate estimate into our Likelihood-based cost function as a single additional term would, without some difficult-to-define regularization, become overwhelmed by the many tracer and variance terms defined in Eqs. (6) and (7). Rather, the additional information is treated as a second objective, namely that global N2 fixation should be greater than  $60 \text{ Tg N yr}^{-1}$  (see above), which is similar to applying a multi-objective approach for model calibration (e.g., Sauerland et al., 2019), where a trade-off between two or more objectives (cost functions) is resolved."
- 2. What is the selling point of this "optimized" flexible C:N:P model? Authors state that most NPZD models do not adequately describe the behavior of plankton physiology such as non-Redfieldian plankton stoichiometry. However, outside the UVic framework, there are quite a few ESMs in the market already with flexible C:N:P including those in CMIP5 (see Bopp et al., 2013) and CMIP6 (see Arora et al., 2019). There are also some studies that utilize Pahlow's phytoplankton model (Kwiatkowski et al., 2018, 2019). My question then is what is the selling point of this model over other existing

models out there? Is it the computational efficiency and how useful is this model for studying climatic conditions such as the last glacial maximum or future projections (lines 39)? I think some discussions on model comparisons would be useful.

**Reply:** The combination of optimality-based nitrogen fixation (Pahlow et al., 2013) and optimal current feeding for zooplankton (Pahlow, 2010), together with the flexible C:N:P stoichiometry are the novel features in the OPEM. Kwiatkowski et al. (2018, 2019) adopted a previous optimality-based model for phytoplankton growth (Pahlow et al., 2009) that does not include optimal resource allocation for nitrogen fixation. It also lacks the optimal current feeding model of zooplankton. None of the biogeochemical modules of the ocean models in CMIP5 and CMIP6 resolve dynamics with respect to the optimality conditions applied in OPEM. PELAGOS (Vichi, Pinardi, and Masina, 2007) is the only model application with variable C:N:P in phytoplankton in CMIP5 (Bopp et al., 2013) and CMIP6 (Arora et al., 2019). It does not consider diazotrophy, and other models resolve either variable N:P (TOPAZ2, Dunne et al., 2013) or variable C:P (MARBL (CESM2), Danabasoglu et al., 2020). In addition to the variable C:N:P stoichiometry, the optimality-based formulations of primary producers and zooplankton have a demonstrated ability to describe processes observed in the laboratory as well as in mesocosm studies and hence provide a strong mechanistic foundation for OPEM. We have added a comparative description in the introduction in this ms on p. 2, lines 51–55: "PELAGOS (Vichi et al., 2007), the only ocean model with variable C:N:P in phytoplankton in CMIP5 (Bopp et al., 2013) and CMIP6 (Arora et al., 2019), has no diazotrophs, others either have only variable N:P (TOPAZ2, Dunne et al., 2013), or variable C:P (MARBL, Danabasoglu et al., 2020). While some of the existing models have a variable C:N:P based on the optimality-based model for phytoplankton growth (Kwiatkowski et al., 2018, 2019), optimality-based  $N_2$  fixation is not included.", as well as extended that in Part I on p. 3, lines 65–68: "We view the implementation of OPEM as one step towards the ultimate goal of reconciling plankton-organism behaviour as observed in the laboratory with global marine biogeochemistry. Therefore, the variable stoichiometry of primary producers should be considered but one, albeit central, aspect of the mechanistic foundation of OPEM." in Pahlow et al. (2020).

3. How sensitive is "sensitive"? Authors discuss the sensitivity of each parameter in Section 3.1 but one thing I find problematic is that all the graphs in Figure 1 – 3 have different y-scale increments. Since sensitivity is non-dimensional, they should ideally all have the same axis for a fair comparison since authors frequently say things like "Sensitivity of XXX is low" (e.g., line 196) or "No single parameter dominates sensitivity" (line 217). Although such rigorous statistical treatments may not be expected for this kind of modeling work, I want some general clarifications on how authors interpreted whether something is very sensitive or not.

**Reply:** The different scales of the y-axes result from our definition of sensitivity in Eq. (3), in which the tracer difference is divided by the average. Some tracers, e.g., DIC or dissolved Fe, vary much less relative to their average concentration, because they are more strongly determined by boundary conditions (air-sea exchange, atmospheric deposition) or exhibit a huge background concentration (DIC) that is, on the timescales considered, not affected by biotic or physical processes. Naturally, we expect that our measure of sensitivity, although normalised and thus being dimensionless, varies on different scales for different tracers or rates. We see no reason for why this may appear problematic, since our focus here is on contrasting the effects of the different parameters and not of the different tracers. In fact, it is important for us to take advantage of the different y-axis scales. Doing so reflects more clearly how sensitive each individual tracer is to variations in the different parameters. The information of the differences between the tracers' general sensitivities to parameter variations is maintained, but according to our chosen style of presentation the emphasis is on the sensitivity to variations in the individual parameters.

We agree with the referee that we need to be more careful with statements like "the sensitivity of X is low", in particular, if we cannot provide a common reference point. It is more appropriate to refer to relative differences in the sensitivities, which we have considered in our corrections. Thus, we now use phrases like

"sensitivity of X is lower than Y". The text of the description of "No single parameter dominates sensitivity" for N2 fixation has been corrected accordingly. The rephrased description on p. 10, lines 238–241, now reads: "The simulated global N2 fixation rate is sensitive to many parameters, apart from  $A_0$  and  $Q_0^P dia$ . Similar relative changes of most parameter values introduce changes to the global N2 fixation rate that are of similar magnitude. Interestingly, N2 fixation is sensitive also to zooplankton parameters, indicating that zooplankton grazing on diazotrophs is an important factor controlling not just diazotroph biomass but also N2 fixation."

4. The highest sensitivity of C:N over C:P and N:P? Regarding the sensitivity, I was quite surprised looking at Figure 3 that C:N has much larger sensitivity compared to C:P and N:P. The current understanding in the scientific community is that C:N is more homeostatic compared to C(N):P for autotrophs, heterotrophs, and for detritus (Galbraith and Martiny, 2015; Geider and La Roche, 2002; Martiny et al., 2013; Sterner and Elser, 2002). Looking at the companion paper by Pahlow et al. (2019), steady-state C:N also seems to overestimate observation (Table 3 and Figure 7). I think this is an important point to address given that C:N (and therefore QoN) affects all aspects of the model output and that the whole point of this model is incorporating flexible C:N:P.

**Reply:** The finding that C:N is more homeostatic than C:P or N:P for particulate matter applies to the spatiotemporal variability in the current ocean and thus could be compared to our trade-off (reference) simulations, which is the topic of Part 1 (Pahlow et al., 2020). The present article, however, describes sensitivities of globally-averaged elemental ratios to parameter variations among 400 simulations with different parameter settings. Thus, the sensitivities discussed here have a very different meaning to the spatio-temporal variability of the elemental ratios in the surface ocean. Particulate C:N and C:P are not only directly affected by  $Q_{0, phv}^{N}$ and  $Q_{0, phy}^{P}$ , but also by the NO3- and PO43- inventories. The marine NO3- inventory varies strongly owing to  $N_2$  fixation and denitrification. In contrast, the  $PO_4^{3-}$  inventory is conserved in the UVic model, allowing only shifts in the spatio-temporal distribution of  $PO_4^{3-}$ . Hence, the sensitivity of globally averaged particulate C:N across simulations with different parameter sets is greater than that of C:P. We have added a discussion to clarify this issue and avoid a possible misunderstanding of the nature of these variations in elemental ratios in the manuscript on p. 22, lines 385–392: "The strong impact of  $Q_{0, phy}^N$  on the NO3- inventory and globally averaged phytoplankton C:N causes a higher sensitivity of globally averaged C:N than C:P (Figure 3). A higher  $Q_{0,vhv}^N$  results in a higher NO3- inventory and a lower phytoplankton C:N, both tending to lower particulate C:N and vice versa. On the other hand, C:P is not as sensitive because we have a constant  $PO_4^{3-}$  inventory in the UVic model. Surface particulate matter C:N is less variable compared to C:P and N:P in field observations along regional gradients (Galbraith and Martiny, 2015; Geider and Roche, 2002; Martiny et al., 2013; Sterner and Elser, 2002), which is an apparent contrast to our results, where the sensitivity of C:N to  $Q_{0, vhy}^{N}$  is the highest among the particulate elemental ratios. However, our sensitivities are with respect to parameter variations among many simulations, rather than spatial or temporal gradients in the one real ocean."

The overestimated steady-state C:N in Part I (Pahlow et al., 2020) results from N2 fixation in the trade-off simulations being much lower than in the current ocean due to the lack of benthic denitrification. Lower N2 fixation results in a lower supply of nitrogen and consequently an overall higher particulate C:N at low latitudes. We have added a discussion on p. 23, lines 432–436: "Also, this means that global N2 fixation (same as global denitrification in our spun-up steady-state simulations) is underestimated, and since it occurs mostly at 40°S to 40°N (see Fig. 13 in Part I, Pahlow et al., 2020), particulate carbon to nitrogen (C:N) ratios could be overestimated due to a missing input of nitrogen to the surface ocean. This could explain the overestimated surface particulate C:N at low latitudes (see Table 3 and Figure 16 in Part I, Pahlow et al., 2020)." We have also added a statement about this topic in Part I (Pahlow et al., 2020) on lines 415–418: "Both the high surface C:N and low P:C in mid-latitude regions might result from the underestimation of N2 fixation, owing to the lack of benthic denitrification. Enhanced N2 fixation would add fixed N to the surface ocean, partly releasing phytoplankton from N limitation and intensifying P limitation, and could thus bring C:N and C:P ratios closer to the observations."

We disagree with the statement that "the whole point of this model is incorporating flexible C:N:P." Flexible C:N:P can be (and has been) implemented in several ways. Although we consider the representation of variable C:N:P in the OPEM very important, our main goal here is, nevertheless, improving the mechanistic foundation of biotic process descriptions in Earth system models.

**Specific comments**

- **Equations:** Diazotrophy rate increases indefinitely with temperature with this formulation. But the growth rate of diazotrophs should hit the limit at some optimal value (e,g., 28 degrees Celsius for Trichodesmium; Breitbarth, E., A. Oschlies, and J. LaRoche (2007), Physiological constraints on the global distribution of Trichodesmium-effect of temperature on diazotrophy, Biogeosciences (BG), 4(1), 53–61). What is the justification of this temperature formulation? I feel like Eppley (1972) is not quite up to date.
- **Reply:** Because there is no equation number indicated, the question is not clear to us. In Equation 1 the rate of N2 fixation indeed increases indefinitely with temperature. While we agree in principle that Eppley (1972) is not quite up to date, this is exactly the reason for introducing Equation 2, which is based on observations, where maximum diazotrophy rate occurs around 25 °C. However, Eppley (1972) is the temperature function in the original UVic, and we wanted to be clear about which changes in model behaviour are due to the optimality-based, variable-stoichiometry formulations, and which are due to the new temperature function. Thus, we set up two model configurations to identify the influence of the temperature dependence of diazotrophy on model behaviour.
- **Line 85:** *The temperature dependence of nitrogenase activity in the terrestrial system was used. Are there not any data from marine ecosystem literature?*
- **Reply:** No, at least we are not aware of any.
- **Table 1:** How are the "Range" chosen for these parameters?
- **Reply:** The parameter ranges are based on literature values. We have revised the description on p. 4, lines 106–108: *"We vary 15 parameters in total, within the variational ranges shown in Table 1, which are based on reference ranges according to literature values."* and added references for the parameters in Table 1.
- Table 2: Maybe it would be nice to have some "target" values for comparison from WOA 2013 or other datasets.
- Reply: We have added the values for the WOA 2013 and other datasets for reference.
- Line 202: What are the sinks for DFe?
- **Reply:** We have a sink for DFe to the sediment, we added this to the text, now on p. 10, line 223: *...iron has a fixed source from atmospheric deposition and a sink in the sediment,...*
- **Figure 2:** *Phytoplankton (1st column) and diazotrophs (2nd column) have different y-axis range. For a fair comparison, they should have the same y range (at least for the same given row).*
- **Reply:** As explained above in our reply to the third general comment, we have decided to keep the y-axis ranges as they were. To avoid confusion, we added *"Note the different y-axis ranges in the different panels."* to the caption.
- **Line 246:** *"their biomass is higher". What is "biomass"? Is it C quota or C+N+P or Chl? I do not see "biomass" in Figure 2.*

- **Reply:** We use the term biomass to refer to C, so higher biomass means higher POC content. The higher diazotroph biomass (Carbon, vertically-integrated and temporally-averaged biomass, mmol C m-2) can be seen in Fig. 15 in Pahlow et al. (2019) (now Pahlow et al., 2020). We revised the sentence to *"is generally larger because of the growth of diazotrophs at high latitudes (see Fig. 15 in Part I, Pahlow et al., 2020)"* on p. 11, lines 264–265 to indicate this. Note that we changed Pahlow et al. (2019) to Pahlow et al. (2020) in the manuscript.
- **Line 254–257:** The logical behind explaining C:P pattern is not clear. Why does NO3:PO4 supply stoichiometry only affect low latitudes? Why that fact P-limitation is not present in S. Ocean explain the negative correlation between C:P and  $Q_{0, vhv}^N$ ?
- **Reply:** To clarify the explanation, we revised this part as: "At low latitudes, the effects of  $Q_{0, phy}^{P}$  are suppressed by variations in phytoplankton C, which is affected by  $Q_{0, phy}^{N}$  and the consequent change in nitrate concentration. Nitrate and phosphate are not limiting in the high-latitude Southern Ocean where, under N- and P-replete conditions, cellular C:P is mainly determined by  $Q_{0, phy}^{P}$  and a higher  $Q_{0, phy}^{P}$  would result in a higher cellular P:C (lower C:P). Therefore, the global C:P of total particulate matter, which is dominated by ordinary phytoplankton, is negatively correlated with  $Q_{0, phy}^{P}$ ." on p. 11, lines 277–281.
- **L282:** The description of "trade-off solutions". I went to Pahlow et al. (2019) but I could not easily locate where the discussion is. Could you direct me specifically to where it is?
- **Reply:** The two calibrated reference simulations in Pahlow et al., (2020) are the "trade-off solutions" in this manuscript. We changed "*in the companion paper Pahlow et al.* (2019)" to "*in Part I (reference simulations in Pahlow et al.,* 2020)" on p. 16, line 306 to avoid such confusion.
- **Figures 6 and 7:** What does "low nitrate" and "high nitrate" mean? I may have missed it but are they different model configurations or are they taken from different oceanographic regions?
- **Reply:** The "low nitrate" and "high nitrate" are the simulations with the lowest and highest globally averaged nitrate concentrations in the OPEM configuration. We revised the description in the caption of Figure 6 as: "*Zonally averaged* NO3- *in the World Ocean Atlas* 2013 (*A*), *the simulations with the lowest and highest* NO3- *inventory* (*B*, *D*), *and the trade-off simulation* (*C*) *in the OPEM configuration. Globally averaged* NO3- *concentrations are shown in each panel. Simulations shown here are marked with solid black and open red triangles in Figure 5. Note that the outputs from OPEM and OPEM-H are very similar and only OPEM results are shown here."* and in the text on p. 17, lines 334–335: "*Figures 6 and 7 show zonally averaged* NO3- *and* O2 *in simulations with the lowest and highest* NO3- *and the trade-off simulation in the OPEM configuration."* . We now show the corresponding simulations in Figure 5. We only show simulations from the OPEM now, because distributions in the OPEM-H are very similar to the OPEM.
- Also Figures 6 and 7: It would be nice to have a zonal average from WOA 2013 for comparison.
- **Reply:** We added zonal averages from the WOA 2013 and removed simulations of the OPEM-H configuration since the distributions are very similar to those of OPEM.
- **Line 381:** N:P of diazotrophs is critically important for determining the outcome of competition between diazotrophs and non-diazotrophs so it should be discussed in more depths here (e.g., Weber and Deutsch, 2012).
- **Reply:** While N:P of diazotrophs was proposed to be very important for determining the outcome of competition between diazotrophs and non-diazotrophs, results of our sensitivity analysis of the OPEM do not support this. In Figure 2 we can see while N:P of diazotrophs is most sensitive to diazotroph subsistence P quota  $(Q_{0, dia}^{P})$ , dizotrophs biomass (carbon) itself is much less sensitive to  $Q_{0, dia}^{P}$  than to  $Q_{0, phy}^{N}$  and to  $Q_{0, dia}^{N}$ . In our view,

the competitive abilities for N and P are more important than the N:P ratio for determining the outcome of such competition. We have added a discussion about how N:P affects competition and N2 fixation of diazotrophs on p. 23, lines 409–417, which now reads: "Diazotroph subsistence N and P quotas  $(Q_{0, dia}^{N} and Q_{0, dia}^{P})$ in general have much less influence on particulate stoichiometry than  $Q_{0, phy}^{N}$  and  $Q_{0, phy}^{P}$  because diazotrophs are much less abundant than ordinary phytoplankton. However, diazotroph biomass (carbon) itself is more sensitive to  $Q_{0, dia}^{N}$  than  $Q_{0, phy}^{N}$ , which shows that the diazotroph subsistence quotas are still important for both their elemental stoichiometry and ability to compete with ordinary phytoplankton. While elemental stoichiometry has been suggested to be an important factor for determining the outcome of the competition between diazotrophs and non-diazotrophs, and consequently N2 fixation (Deutsch and Weber, 2012; Weber and Deutsch, 2012), we find that N2 fixation is no more sensitive to  $Q_{0, dia}^{N}$ than to the remineralisation rate ( $\nu_{det}$ ),  $Q_{0, phy}^{N}$ , or zooplankton grazing parameters ( $g_{max}$ ,  $\phi_{phy}$ , and  $\phi_{dia}$ ). Nevertheless, our analysis agrees with the argument that global N2 fixation is mainly determined by rates of fixed-N loss (Weber and Deutsch, 2014), which in our model is largely affected by  $\nu_{det}$  and  $Q_{0, phy}^{N}$ ."

- **Line 407:** I think authors should also mention the fact that physical component/ocean circulation is very important for the global distribution of oxygen and nitrate.
- **Reply:** We have added a statement that the physical component/ocean circulation is very important for the global distribution of oxygen and nitrate. Now on p. 24, lines 450–451: "While the physical component (ocean circulation) of the UVic model is also very important for the global distribution of oxygen and nitrate, our results suggest that..."

 Table 1. Parameter names, reference and variational ranges, identified "best" values for the trade-off simulations in (OPEM and OPEM-H), units and descriptions. Note that the trade-off simulations share the same parameter combination.

| Symbol                                        | Range-Reference                 | Variational
range | OPEM/ OPEM-H | Units                                       | Definition                                    |
|-----------------------------------------------|---------------------------------|----------------------|--------------|---------------------------------------------|-----------------------------------------------|
| $A_{0, \mathrm{phy}}$                         | 70–1000a      | 120-280              | 229          | $m^{3} (mol C)^{-1} d^{-1}$                 | phytoplankton potential nutrient affinity     |
| $Q_{0, \mathrm{ phy}}^{\mathrm{N}}$           | 0.038-0.086 a        | 0.04-0.06            | 0.04128      | $\mathrm{mol}(\mathrm{mol}\mathrm{C})^{-1}$ | phytoplankton subsistence N quota             |
| $Q_{0,\mathrm{dia}}^{\mathrm{N}}$             | 0.13 a        | 0.06-0.12            | 0.067        | $\mathrm{mol}(\mathrm{mol}\mathrm{C})^{-1}$ | diazotroph subsistence N quota                |
| $Q^{ m P}_{0, \  m phy}$                      | 0.0008-0.002 a       | 0.0013-0.0023        | 0.0022       | $\mathrm{mol}(\mathrm{mol}\mathrm{C})^{-1}$ | phytoplankton subsistence P quota             |
| $Q^{ m P}_{0,~ m dia}$                        | 0.0027 a             | 0.0025-0.0035        | 0.00271      | $\mathrm{mol}(\mathrm{mol}\mathrm{C})^{-1}$ | diazotroph subsistence P quota                |
| $k_{ m Fe,\ phy}$                             | 0.035-0.12 c-g       | 0.04-0.08            | 0.066        | $\mu molm^{-3}$                             | phytoplankton half-saturation constant for Fe |
| $g_{ m max}$                                  | 0.49–5 a             | 1–2                  | 1.75         | $d^{-1}$                                    | zooplankton maximum specific ingestion rate   |
| $\phi_{ m phy}$                               | $\underbrace{174-765^{h}}_{}$   | 100-200              | 118          | $\rm m^3(molC)^{-1}$                        | capture coefficient of phytoplankton          |
| $\phi_{ m dia}$                               | $1.05 \cdot \phi_{phy}$         | 150-250              | 232          | $\rm m^3(molC)^{-1}$                        | capture coefficient of diazotrophs            |
| $\phi_{ m det}$                               | $\phi_{\rm phy} \sim^{\rm c-f}$ | 20-100               | 94           | $m^3(molC)^{-1}$                            | capture coefficient of detritus               |
| $\phi_{\sf zoo}$                              | $\underline{0-3230^{h}}$        | 100-200              | 118          | $m^3(molC)^{-1}$                            | capture coefficient of zooplankton            |
| $\lambda_{0, 	ext{ phy}} = M_{0, 	ext{ dia}}$ | $0.001-0.015^{c-f}$             | 0.01-0.03            | 0.018        | $d^{-1}$                                    | specific mortality rate                       |
| $ u_{ m det}$                                 | 0.05-0.07 c-g        | 0.04-0.09            | 0.087        | $d^{-1}$                                    | remineralization rate                         |

a(Pahlow, 2005; Pahlow et al., 2013), b(Pahlow and Prowe, 2010), c(Keller et al., 2012), d(Somes and Oschlies, 2015), e(Somes et al., 2017) f(Landolfi et al., 2017), g(Landolfi et al., 2015), h(Su et al., 2018), i(Wang et al., 2019)

observations and model results:

$$\left(\overline{\log_{10} X}\right)_{jk} = \frac{1}{N_{jk}} \sum_{n=1}^{N_{jk}} \left(\underline{\ln \log_{10}}_{\infty \infty} \left[\frac{\max(X_{(n)}, X_{(0)})}{X_{(0)}}\right]\right), \qquad X \in \{\text{Chl}, O_2, \text{NO}_3^-, \text{PO}_4^{3-}\}$$
(4)

- 155 where  $N_{jk}$  is the number of available data points within biome j in depth level k. Prior to log-transformation the log10-transformation, all tracer concentrations have been normalised to lower detection (uncertainty) thresholds  $(X_{(0)})$  respectively. Measured or derived concentrations below these thresholds are treated as noise and therefore remain unresolved. Thus, the log-transformed log10-transformed normalised concentrations are non-negative. The threshold-values are:  $Chl_{(0)} = 0.1 \text{ mg m}^{-3}$ ,  $O_{2(0)} = 1 \text{ mmol m}^{-3}$ ,  $NO_3^{-}_{(0)} = 0.05 \text{ mmol m}^{-3}$ , and  $PO_4^{3-}_{(0)} = 0.01 \text{ mmol m}^{-3}$ .
- 160 Our metric is derived from a likelihood, assuming a Gaussian error distribution for the residuals, which describe the discrepancy between mean values derived from observations  $(\frac{\ln X^{(obs)} \log_{10} X^{(obs)}}{\log_{10} X^{(obs)}})$  and model simulations  $(\frac{\ln X^{(mod)} \log_{10} X^{(mod)}}{\log_{10} X^{(mod)}})$ .

Hereafter we refer to this metric as our cost function (J). Our cost function is split up into two major parts:

170

$$J = \sum_{k=1}^{5} J_k^{(u)} + \sum_{k=6}^{19} J_k^{(l)}$$
(5)

$$J_{k}^{(u)} = \sum_{i=1}^{12} \sum_{j=1}^{17} \left[ \mathbf{d}^{T} \ R^{-1} \ \mathbf{d} \right]_{ijk} + \left( \mathbf{v}^{(\text{obs})} - \mathbf{v}^{(\text{mod})} \right)_{ijk}^{T} V_{ijk}^{-1} \left( \mathbf{v}^{(\text{obs})} - \mathbf{v}^{(\text{mod})} \right)_{ijk}$$
(6)

165
$$J_{k}^{(l)} = \sum_{j=1}^{1} \left[ \mathbf{d}^{T} R^{-1} \mathbf{d} \right]_{jk} + \left( \mathbf{v}^{(\text{obs})} - \mathbf{v}^{(\text{mod})} \right)_{jk}^{T} V_{jk}^{-1} \left( \mathbf{v}^{(\text{obs})} - \mathbf{v}^{(\text{mod})} \right)_{jk}$$
(7)

[revised manuscript text omitted]

| Tracer                  | OPEM                              | OPEM-H                             | Reference                                                                                                                                                                                                                                                                                                                                                                                                                                                                                                                                                                                                                                                                                                                                                                                                                                                                                                                                                                                                                                                                                                                                                                                                                                                                                                                                                                                                                                                                                                                                                                                                                                                                                                                                                                                                                                                                                                                                                                                                                                                                                                                                                                                                                                                                                                                                                                                                                                                                                                                                                                                                                                                                                                                                                                                                                                                                                                                                                                                                                                    | Units                          |
|-------------------------|-----------------------------------|------------------------------------|----------------------------------------------------------------------------------------------------------------------------------------------------------------------------------------------------------------------------------------------------------------------------------------------------------------------------------------------------------------------------------------------------------------------------------------------------------------------------------------------------------------------------------------------------------------------------------------------------------------------------------------------------------------------------------------------------------------------------------------------------------------------------------------------------------------------------------------------------------------------------------------------------------------------------------------------------------------------------------------------------------------------------------------------------------------------------------------------------------------------------------------------------------------------------------------------------------------------------------------------------------------------------------------------------------------------------------------------------------------------------------------------------------------------------------------------------------------------------------------------------------------------------------------------------------------------------------------------------------------------------------------------------------------------------------------------------------------------------------------------------------------------------------------------------------------------------------------------------------------------------------------------------------------------------------------------------------------------------------------------------------------------------------------------------------------------------------------------------------------------------------------------------------------------------------------------------------------------------------------------------------------------------------------------------------------------------------------------------------------------------------------------------------------------------------------------------------------------------------------------------------------------------------------------------------------------------------------------------------------------------------------------------------------------------------------------------------------------------------------------------------------------------------------------------------------------------------------------------------------------------------------------------------------------------------------------------------------------------------------------------------------------------------------------|--------------------------------|
| Oxygen                  | 99.6–219                          | 103–214                            | $176^{a}$                                                                                                                                                                                                                                                                                                                                                                                                                                                                                                                                                                                                                                                                                                                                                                                                                                                                                                                                                                                                                                                                                                                                                                                                                                                                                                                                                                                                                                                                                                                                                                                                                                                                                                                                                                                                                                                                                                                                                                                                                                                                                                                                                                                                                                                                                                                                                                                                                                                                                                                                                                                                                                                                                                                                                                                                                                                                                                                                                                                                                                    | ${\rm mmolm^{-3}}$             |
| Nitrate                 | 10.2–66.2                         | 13.0–55.0                          | 31 b                                                                                                                                                                                                                                                                                                                                                                                                                                                                                                                                                                                                                                                                                                                                                                                                                                                                                                                                                                                                                                                                                                                                                                                                                                                                                                                                                                                                                                                                                                                                                                                                                                                                                                                                                                                                                                                                                                                                                                                                                                                                                                                                                                                                                                                                                                                                                                                                                                                                                                                                                                                                                                                                                                                                                                                                                                                                                                                                                                                                                              | ${\rm mmolm^{-3}}$             |
| DIC                     | 2.239-2.439                       | 2.248-2.430                        | 2.317 c                                                                                                                                                                                                                                                                                                                                                                                                                                                                                                                                                                                                                                                                                                                                                                                                                                                                                                                                                                                                                                                                                                                                                                                                                                                                                                                                                                                                                                                                                                                                                                                                                                                                                                                                                                                                                                                                                                                                                                                                                                                                                                                                                                                                                                                                                                                                                                                                                                                                                                                                                                                                                                                                                                                                                                                                                                                                                                                                                                                                                           | $ m molm^{-3}$                 |
| DFe                     | 0.47-0.71                         | 0.47-0.69                          | $\underbrace{0.57^d}_{\overset{}\overset{}\overset{}\overset{}\overset{}\overset{}\overset{}\overset{}\overset{}\overset{}\overset{}\overset{}\overset{}\overset{}\overset{}\overset{}\overset{}\overset{}\overset{}\overset{}\overset{}\overset{}\overset{}\overset{}\overset{}\overset{}\overset{}\overset{}\overset{}\overset{}\overset{}\overset{}\overset{}\overset{}\overset{}\overset{}\overset{}\overset{}\overset{}\overset{}\overset{}\overset{}\overset{}\overset{}\overset{}\overset{}\overset{}\overset{}\overset{}\overset{}\overset{}\overset{}\overset{}\overset{}\overset{}\overset{}\overset{}\overset{}\overset{}\overset{}\overset{}\overset{}\overset{}\overset{}\overset{}\overset{}\overset{}\overset{}\overset{}\overset{}\overset{}\overset{}\overset{}\overset{}\overset{}\overset{}\overset{}\overset{}\overset{}\overset{}\overset{}\overset{}\overset{}\overset{}\overset{}\overset{}\overset{}\overset{}\overset{}\overset{}\overset{}\overset{}\overset{}\overset{}\overset{}\overset{}\overset{}\overset{}\overset{}\overset{}\overset{}\overset{}\overset{}\overset{}\overset{}\overset{}\overset{}\overset{}\overset{}\overset{}\overset{}\overset{}\overset{}\overset{}\overset{}\overset{}\overset{}\overset{}\overset{}\overset{}\overset{}\overset{}\overset{}\overset{}\overset{}\overset{}\overset{}\overset{}\overset{}\overset{}\overset{}\overset{}\overset{}\overset{}\overset{}\overset{}\overset{}\overset{}\overset{}\overset{}\overset{}\overset{}\overset{}\overset{}\overset{}\overset{}\overset{}\overset{}\overset{}\overset{}\overset{}\overset{}\overset{}\overset{}\overset{}\overset{}\overset{}\overset{}\overset{}\overset{}\overset{}\overset{}\overset{}\overset{}\overset{}\overset{}\overset{}\overset{}\overset{}\overset{}\overset{}\overset{}\overset{}\overset{}\overset{}\overset{}\overset{}\overset{}\overset{}\overset{}\overset{}\overset{}\overset{}\overset{}\overset{}\overset{}\overset{}\overset{}\overset{}\overset{}\overset{}\overset{}\overset{}\overset{}\overset{}\overset{}\overset{}\overset{}\overset{}\overset{}\overset{}\overset{}\overset{}\overset{}\overset{}\overset{}\overset{}\overset{}\overset{}\overset{}\overset{}\overset{}\overset{}\overset{}\overset{}\overset{}\overset{}\overset{}\overset{}\overset{}\overset{}\overset{}\overset{}\overset{}\overset{}\overset{}\overset{}\overset{}\overset{}\overset{}\overset{}\overset{}\overset{}\overset{}\overset{}\overset{}\overset{}\overset{}\overset{}\overset{}\overset{}\overset{}\overset{}\overset{}\overset{}\overset{}\overset{}\overset{}\overset{}\overset{}\overset{}\overset{}\overset{}\overset{}\overset{}\overset{}\overset{}\overset{}\overset{}\overset{}\overset{}\overset{}\overset{}\overset{}\overset{}\overset{}\overset{}\overset{}\overset{}\overset{}\overset{}\overset{}\overset{}\overset{}\overset{}\overset{}\overset{}\overset{}\overset{}\overset{}\overset{}\overset{}\overset{}\overset{}\overset{}}\overset{}\overset{}\overset{}\overset{}\overset{}$ | $\mu {\rm mol}{\rm m}^{-3}$    |
| PFe                     | 0.44-0.75                         | 0.44-0.70                          | $\underbrace{1.17^d}_{\overset{d}{}}$                                                                                                                                                                                                                                                                                                                                                                                                                                                                                                                                                                                                                                                                                                                                                                                                                                                                                                                                                                                                                                                                                                                                                                                                                                                                                                                                                                                                                                                                                                                                                                                                                                                                                                                                                                                                                                                                                                                                                                                                                                                                                                                                                                                                                                                                                                                                                                                                                                                                                                                                                                                                                                                                                                                                                                                                                                                                                                                                                                                                        | $\mathrm{nmol}\mathrm{m}^{-3}$ |
| Chl                     | <del>37.6–101.2</del> 0.123–0.332 | <del>38.0-103.5</del> -0.128-0.336 | 0.309 e                                                                                                                                                                                                                                                                                                                                                                                                                                                                                                                                                                                                                                                                                                                                                                                                                                                                                                                                                                                                                                                                                                                                                                                                                                                                                                                                                                                                                                                                                                                                                                                                                                                                                                                                                                                                                                                                                                                                                                                                                                                                                                                                                                                                                                                                                                                                                                                                                                                                                                                                                                                                                                                                                                                                                                                                                                                                                                                                                                                                                           | $\mathrm{mgm}^{-2}$            |
| NPP                     | 27.8-88.0                         | 27.2-88.0                          | $52^{f}$                                                                                                                                                                                                                                                                                                                                                                                                                                                                                                                                                                                                                                                                                                                                                                                                                                                                                                                                                                                                                                                                                                                                                                                                                                                                                                                                                                                                                                                                                                                                                                                                                                                                                                                                                                                                                                                                                                                                                                                                                                                                                                                                                                                                                                                                                                                                                                                                                                                                                                                                                                                                                                                                                                                                                                                                                                                                                                                                                                                                                                     | ${\rm Pg}{\rm Cyr}^{-1}$       |
| NCP                     | <del>0.86-3.01</del> 8.0-16.4     | <del>0.79-3.20</del> 7.8-16.2      | $\underbrace{13.5^g}$                                                                                                                                                                                                                                                                                                                                                                                                                                                                                                                                                                                                                                                                                                                                                                                                                                                                                                                                                                                                                                                                                                                                                                                                                                                                                                                                                                                                                                                                                                                                                                                                                                                                                                                                                                                                                                                                                                                                                                                                                                                                                                                                                                                                                                                                                                                                                                                                                                                                                                                                                                                                                                                                                                                                                                                                                                                                                                                                                                                                                        | ${\rm Pg}{\rm Cyr}^{-1}$       |
| N 2 Fixation | 0–480                             | 0–518                              | $140^{h}$                                                                                                                                                                                                                                                                                                                                                                                                                                                                                                                                                                                                                                                                                                                                                                                                                                                                                                                                                                                                                                                                                                                                                                                                                                                                                                                                                                                                                                                                                                                                                                                                                                                                                                                                                                                                                                                                                                                                                                                                                                                                                                                                                                                                                                                                                                                                                                                                                                                                                                                                                                                                                                                                                                                                                                                                                                                                                                                                                                                                                                    | ${ m Tg}{ m N}{ m yr}^{-1}$    |

aWOA 2013 (Garcia et al., 2013a)

bWOA 2013 (Garcia et al., 2013b)

cGLODAPv2 (Olsen et al., 2016)

d(Nickelsen et al., 2015),

eMODIS/Aqua level 3, 2008–2017 (Ocean Biology Processing Group, 2014)

f(Westberry et al., 2008)

g(Li and Cassar, 2016)

h(Luo et al., 2012)

**3.1 Sensitivity to Model Parameters**

**3.1.1 Biogeochemical tracer inventories and governing processes**

The sensitivities of globally averaged biogeochemical properties to the variations of each of the 13 parameters in Table 2 are comparable for OPEM and OPEM-H (Figure 1). Global mean oxygen concentration is most sensitive to  $\nu_{det}$  (remineralization

- 205 rate). Higher  $\nu_{det}$  increases oxygen consumption in shallow water, where oxygen resupply from the atmosphere is stronger. Less oxygen is consumed below the surface ocean, hence the total oxygen inventory increases. Maximum ingestion rate  $(g_{max})$  and grazing rate on ordinary phytoplankton  $(\phi_{phy})$  also correlate positively with oxygen. Higher  $g_{max}$  or  $\phi_{phy}$  means more ordinary phytoplankton is grazed and less particles are formed, which then decreases oxygen consumption through remineralization. Oxygen is less sensitive to  $\phi_{dia}$ , because the biomass of diazotrophs is much smaller than that of ordinary phytoplankton.
- A surprising finding is that oxygen is sensitive to, and positively correlated with, the subsistence nitrogen quota of ordinary phytoplankton  $(Q_{0, phy}^{N})$ . From a classic point of view, oxygen levels in the ocean are dominated by physical supply processes as well as biogeochemical consumption processes such as remineralization (Feely et al., 2004). Nevertheless, in our simulations the sensitivity to  $Q_{0, phy}^{N}$  is more than half (58%) of that to  $\nu_{det}$  in OPEM and 48% in OPEM-H (Figure 1). In our model,  $Q_{0, phy}^{N}$ has no effect on the spatial distribution of cellular C:N ratios in phytoplankton, which is determined by ambient light and nutrient conditions. However,  $Q_{0, phy}^{N}$  affects the average phytoplankton C:N ratio. The average phytoplankton C:N ratio decreases
- when Q0, phyN increases, with less carbon being fixed for the same NO3- supply. Oxygen consumption (due to remineralization) per mole of nitrogen thus decreases in consequence. Q0, phyN in turn affects NO3-: A higher Q0, phyN yields a higher oxygen level and hence less denitrification in oxygen deficient zones (ODZs) and therefore leads to more NO3-. In fact, we identify this as a major process that controls the NO3- inventory in our simulations (Figure 1). While NO3- is also sensitive to other parameters,
  220 its sensitivity to Q0, phyN is more than twice that to any other parameter (Figure 1).

The sensitivity of dissolved inorganic carbon (DIC) is generally low, because of the relatively large DIC pool compared to the variations in fluxes among the different parameter sets. Similar to oxygen, DIC is most sensitive to  $\nu_{det}$ ,  $Q_{0, phy}^{N}$ ,  $g_{max}$  and  $\phi_{phy}$ . Faster carbon recycling in the surface layer due to higher  $\nu_{det}$  generates a higher surface DIC concentration and hence more outgassing, which decreases the DIC inventory. A somewhat lower DIC inventory is also induced by a larger  $Q_{0, phy}^{N}$ , as less carbon is fixed and exported per unit nitrogen in phytoplankton, and by enhanced zooplankton grazing with larger  $g_{max}$ .

Dissolved iron (DFe) is most sensitive to the remineralisation rate (vdet). Unlike NO3-, which has dynamic source (N2 fixation) and sink (denitrification) processes, iron has a fixed source from atmospheric deposition and the a sink in the sediment, and the size of the DFe pool is mainly determined by its internal cycle. A higher remineralisation rate prolongs the residence time and thus increases the DFe pool. The parameter vdet also indirectly affects the internal DFe cycle via its effect on O2.
While the detritus remineralisation rate drops when O2 falls below 5 mmolm-3 (Nickelsen et al., 2015), scavenging of DFe stops below the same oxygen threshold. Detritus remineralisation rate dominates variations in DFe when globally averaged

 $O_2$  is above 135 mmol m-3, in which case DFe is positively correlated with  $\nu_{det}$  and  $O_2$ . When globally averaged  $O_2$  is below 135 mmol m-3, the wide-spread ODZs (below 5 mmol m-3) inhibit the scavenging of DFe and this effect dominates. As a result, DFe becomes anti-correlated with  $O_2$ . Particulate iron (PFe) is also positively correlated with  $\nu_{det}$  when globally